# Gravity compensation for leachate grid cleaning robots in waste-to-energy plants: A modeling and simulation study

Angang Cao 🄳[1]*, Cong Wang[2], Wei Li[1], Hongwen Ma[2]

**1** School of Mechanical Engineering, Zhengzhou University of Science and Technology, Zhengzhou, China, **2** School of of Mechanical and Electrical Engineering, Harbin Engineering University, Harbin, China

* caoangang88@163.com

## Abstract

Leachate grid clogging in waste-to-energy plants severely reduces combustion efficiency by up to 42% and may cause unplanned shutdowns, leading to substantial economic losses, necessitating automated cleaning solutions. However, gravity-induced elastic deformation in long-reach hydraulic manipulators limits their positioning precision. This research introduces an innovative feedforward static compensation strategy for hydraulically driven weak-rigid manipulators, which leverages detailed analytical joint stiffness modeling and avoids the inherent latency of conventional feedback-based methods (relying on force sensors or parameter identification); it is validated via comprehensive Adams multibody dynamics simulations. Validated via Adams multibody dynamics simulations, the proposed method reduced the end-effector positioning errors by 97.49% (X), 92.91% (Y), and 94.84% (Z), achieving a repeatable positioning accuracy of ±0.1 mm—far exceeding the 0.5 mm requirement for automated grid cleaning. This model-based compensation strategy provides a generalizable theoretical framework for precision control of long-reach hydraulic manipulators. The current study is validated through high-fidelity Adams simulations, achieving a simulated repeatable positioning accuracy of ±0.1 mm. However, it is essential to emphasize that these results represent an idealized upper bound. Real-world hydraulic systems introduce complexities not captured in simulation—including backlash, nonlinear friction, valve dynamics, hysteresis, leakage, and sensor quantization—all of which will degrade practical accuracy. Therefore, this work establishes a theoretical foundation requiring experimental validation on physical hardware. Future work will focus on physical prototyping and on-site testing to address real-world challenges.

**Data availability statement:** All relevant data are within the paper and its Supporting Information files.

**Funding:** This work was supported by the Henan Province Science and Technology Research Project(No.252102220033), Natural Science Foundation of Henan（No.252300420072）and the education department of Henan Province (No. 24A460023 and No. 26A460029).

**Competing interests:** The authors declare that they have no known competing financial interests or personal relationships that could have appeared to influence the work reported in this paper.

## 1. Introduction

Confronted with the escalating global waste crisis, cities worldwide are increasingly turning to waste-to-energy technologies as a sustainable solution for municipal solid waste management, with a notable surge in installations over the past decade [1–3]. However, untreated municipal solid waste typically exhibits high moisture content; direct combustion can lead to calorific value losses of up to 42%, significantly diminishing power generation efficiency [4–6]. To mitigate this, waste is stored in pits for dehydration and fermentation. The liquid extracted during this process, termed "leachate," drains through perforated grids (grid filter screens) installed on the pit walls (Fig 1) [7,8].

However, due to the complex composition and varying particle sizes of waste, high viscosity of leachate, and relatively narrow grid apertures, grid clogging occurs readily [9–12], as shown in Fig 2. Grid blockage obstructs leachate drainage, leading to liquid accumulation within the waste pit. This not only drastically reduces waste incineration efficiency but can also necessitate emergency shutdowns for manual unclogging, resulting in substantial economic losses. Therefore, achieving efficient, reliable, and automated unclogging of leachate grids is a critical technological bottleneck for ensuring the stable operation and economic viability of waste-to-energy plants.

To prevent production disruptions caused by clogged grid apertures, regular and timely cleaning is essential. Manual cleaning methods are commonly employed but exhibit significant drawbacks: (1) Numerous cleaning windows lead to high labor intensity; (2) Toxic and hazardous gases generated during waste fermentation pose severe safety risks; (3) The environment within the leachate corridor is characterized by foul odors, heat, and poor air circulation, creating unfavorable working conditions.

Currently, several leachate grid cleaning devices exist, primarily including manipulator-type high-pressure water jet robots, Cartesian coordinate cleaning devices, built-in unclogging mechanisms, and wheeled mobile unclogging robots [13–16]. A comparison of existing devices is presented in Table 1.

Analysis of existing technologies reveals distinct limitations in effectively balancing stability, safety, and cost-effectiveness for the demanding leachate grid environment. Fixed Cartesian coordinate robots and built-in mechanisms, while simple in principle, require deployment at each grid unit, leading to prohibitively high installation and maintenance costs [13,15]. Furthermore, their inherent rigidity makes them incapable of adapting to the inevitable grid deformation caused by waste impact, risking equipment damage or further grid perforation [14]. Wheeled mobile robots offer conceptual flexibility but struggle with insufficient unclogging force and cannot maintain positional stability under the reaction forces during drilling [16].

In contrast, manipulator-type robots present a promising solution due to their high equipment utilization and flexibility, enabling full-layer coverage with a single device. However, a critical gap remains in the precise control of these manipulators, specifically in compensating for gravity-induced elastic deformation in their weakly rigid, long-reach arms. This deformation is the primary obstacle to achieving the sub-millimeter precision required for aligning with deformed grid apertures. Previous studies on gravity compensation for industrial robots have predominantly relied on

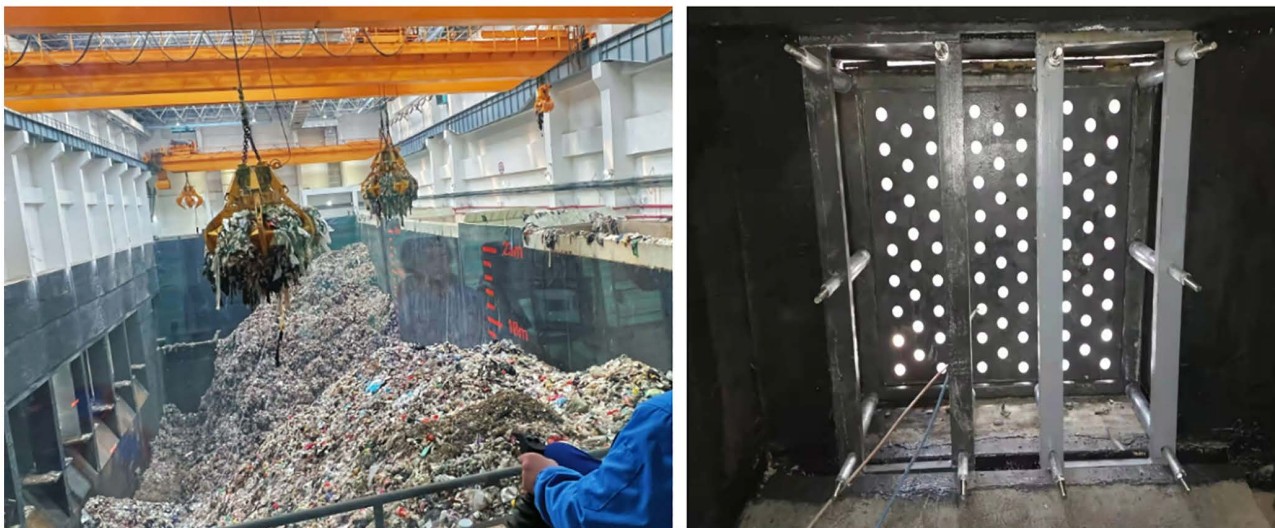

**Fig 1. Waste pit and leachate grid in a waste-to-energy plant.** (a) Waste Storage Pit, (b) Leachate Grid.

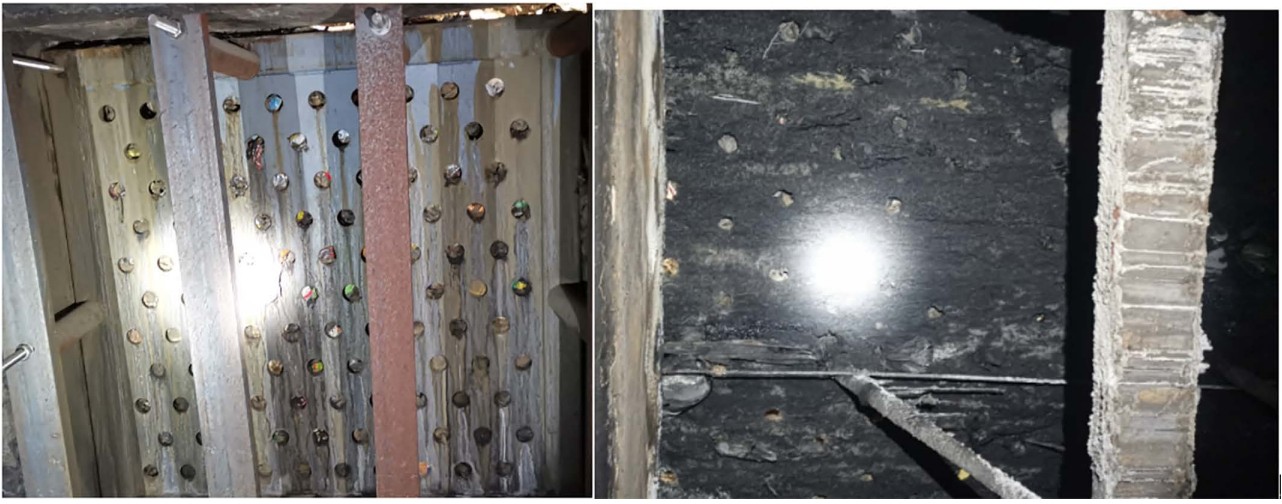

**Fig 2. Clogged Leachate Grid.** (a) Grid with Normal Fluid Passage, (b) Clogged Grid.

feedback strategies based on force sensors or parameter identification. These methods, while effective in many scenarios, are susceptible to system latency—additionally, force sensor-based methods fail in toxic, humid, and dusty leachate corridors, and parameter identification requires frequent calibration (impractical for continuous on-site operations)—making them suboptimal for this application [17–20].

Beyond these feedback-based approaches, significant research has addressed model-based compensation for compliant robots. Studies on quasi-static deflection analysis reveal that traditional static calibration methods can underestimate positioning errors by neglecting combined load and motion effects [21]. Model-based compliance compensation

**Table 1. Comparison of existing leachate grid cleaning devices.**

| Type | Description | Limitations |
|---|---|---|
| High-pressure water manipulator robot | Installed on the lowest corridor level, equipped with rails for mobility. Uses high-pressure water jets at the manipulator end to flush clogged areas | High-pressure water jets only break through the surface layer of clogs, failing to penetrate deeply; poor actual unclogging effectiveness |
| Cartesian coordinate robot | The robot end-effector consists of multiple unclogging rods matching the grid aperture positions. Simple design, high efficiency during push-pull operation | Requires large quantities of devices, high capital investment, and significant maintenance; grid deformation under waste impact misaligns apertures, rods can damage grid |
| Built-in unclogging mechanism | Replaces the grid with a perforated box structure capable of pushing/pulling actions, allowing leachate to flow through its apertures | Subjected to immense pressure from waste gravity and impact forces during dumping/grab operations; highly prone to damage |
| Wheeled intelligent robot | Employs a wheeled mobile platform on the corridor deck, coupled with a vision system for positioning, unclogging apertures individually | Demanding environmental requirements (clean, flat corridor deck); insufficient unclogging force; cannot maintain position under reaction forces |

incorporating joint stiffness uncertainties has been demonstrated, with sensitivity analyses quantifying the impact of parameter variations on positioning accuracy [22,23]. For machining applications, hybrid compensation schemes combining offline simulation with online force sensing have achieved path accuracy improvements of over 80% [24].

In the hydraulic domain, control uncertainties present additional challenges. Hydraulic systems exhibit complex nonlinearities including valve deadband, pressure dynamics, oil compressibility, and temperature-dependent behavior [25–27]. Adaptive robust control approaches have been developed to compensate for these uncertainties, with displacement dynamics compensation improving tracking accuracy by nearly 20% over conventional methods [26]. Studies on hydraulic construction robots have specifically addressed gravity compensation for teleoperation, demonstrating the feasibility of compensating for object mass during conveyance tasks [28].

Long-reach manipulator applications—including space cranes, boom arms, and excavator-type robots—have particular relevance to our work. Research on these systems has highlighted the dominance of joint compliance over link flexibility, the importance of configuration-dependent gravitational effects, and the practical challenges of achieving sub-millimeter accuracy with hydraulic actuation [29–31]. However, existing studies typically focus on either static compensation without sensitivity analysis, or dynamic control without the specialized joint stiffness modeling required for weak-rigid manipulators in hazardous environments.

This study addresses the specific gap in hydraulically driven weak-rigid manipulators operating in hazardous environments, where existing model-based compensation methods [21–24] have not been tailored to the combination of: (1) explosion-proof hydraulic actuation with significant joint compliance, (2) the need for static feedforward compensation without real-time feedback (due to sensor limitations in toxic/humid environments), and (3) the sub-millimeter accuracy requirement for grid cleaning. Our contribution lies not in proposing a fundamentally new class of compensation, but in integrating analytical joint stiffness modeling with configuration-specific torque calculations for this unique application context, and in providing the first systematic sensitivity analysis of such an approach for leachate corridor robots. This model preemptively calculates and corrects for gravitational errors in the control command, effectively circumventing the latency inherent in closed-loop control.

This work makes three core contributions: 1) Developed a hydraulically driven 5-DOF rail-mounted cleaning robot tailored for hazardous leachate corridors (explosion-proof, adaptive to deformed grids); 2) Proposed a feedforward static gravity compensation strategy based on reducer stiffness modeling, preemptively correcting joint deformation without real-time feedback; 3) Theoretically validated the strategy via Adams simulations, achieving ±0.1 mm repeatable accuracy (exceeding the 0.5 mm engineering requirement) and providing a robust foundation for real-world deployment.

## 2. Robot design and modeling

### 2.1. System requirements

The cleaning robot must operate within leachate corridors (Fig 3), characterized by hazardous conditions: confined space, poor natural ventilation, presence of toxic/hazardous and flammable/explosive gases (e.g., sulfur oxides, carbon oxides) resulting from waste fermentation. Fire and electricity use should be minimized. Key operational constraints include:

(1) Hydraulic Actuation: Essential for explosion-proof operation in hazardous environments. Compared to electric actuation, hydraulic systems offer higher load capacity (≥200 N push force required for unclogging) and better stability in high-temperature, humid conditions.

(2) Single-Probe Cleaning: Necessary to accommodate grid deformations; multi-probe designs risk collision.

(3) Modular Rail-Mounted Mobility: Required for full coverage of the lower grid layer using a single robot unit.

(4) Harsh Environment: Grids are subject to deformation and damage from waste dumping and grabber impacts. Manual clearing typically requires forces below 200N. Leachate primarily drains from the lower grid layer.

### 2.2. Mechanical design

Based on the system requirements, a rail-mounted manipulator robot with hydraulic actuation was designed as the optimal solution (Fig 4). Key features include:

(1) 5-DOF Manipulator: The configuration (3 positioning DOFs + 2 orientation DOFs) is optimized to cover the entire lower grid layer (26 windows) with minimal redundancy, reducing structural weight and gravity-induced deformation while enabling pose adjustment for deformed grids. Joints 1–4 utilize hydraulic motors coupled with rotary reducers. Joint 5, requiring minimal torque, uses a hydraulic motor for direct drive. The end-effector is driven by a hydraulic motor and RV reducer. All geometric parameters (link lengths, joint offsets, and component dimensions) were obtained directly

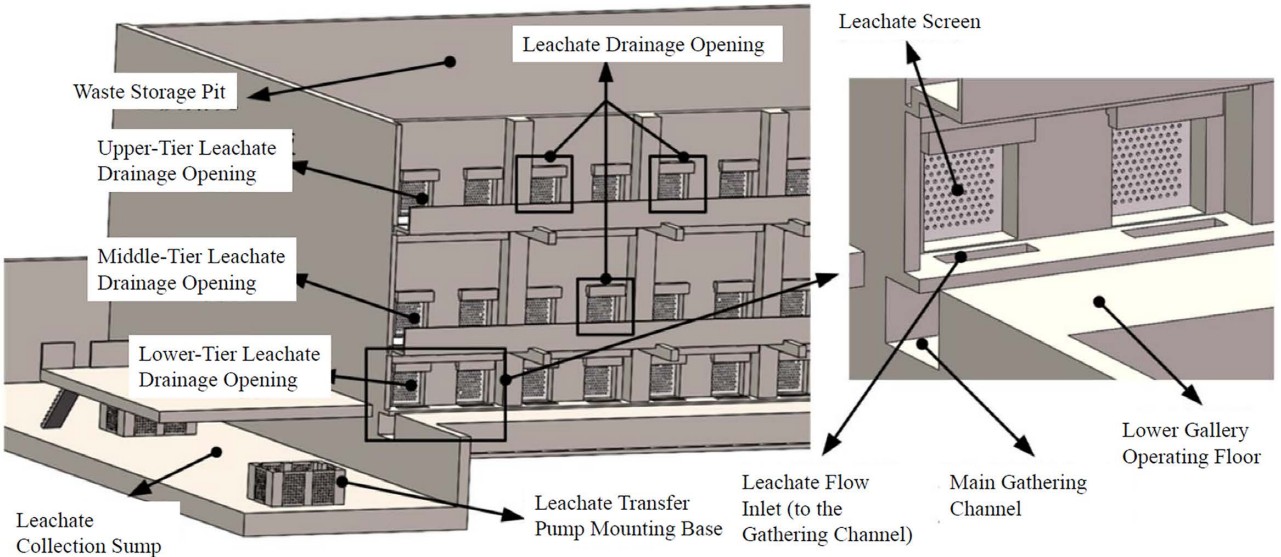

**Fig 3. Leachate corridor structure in a waste incineration plant.**

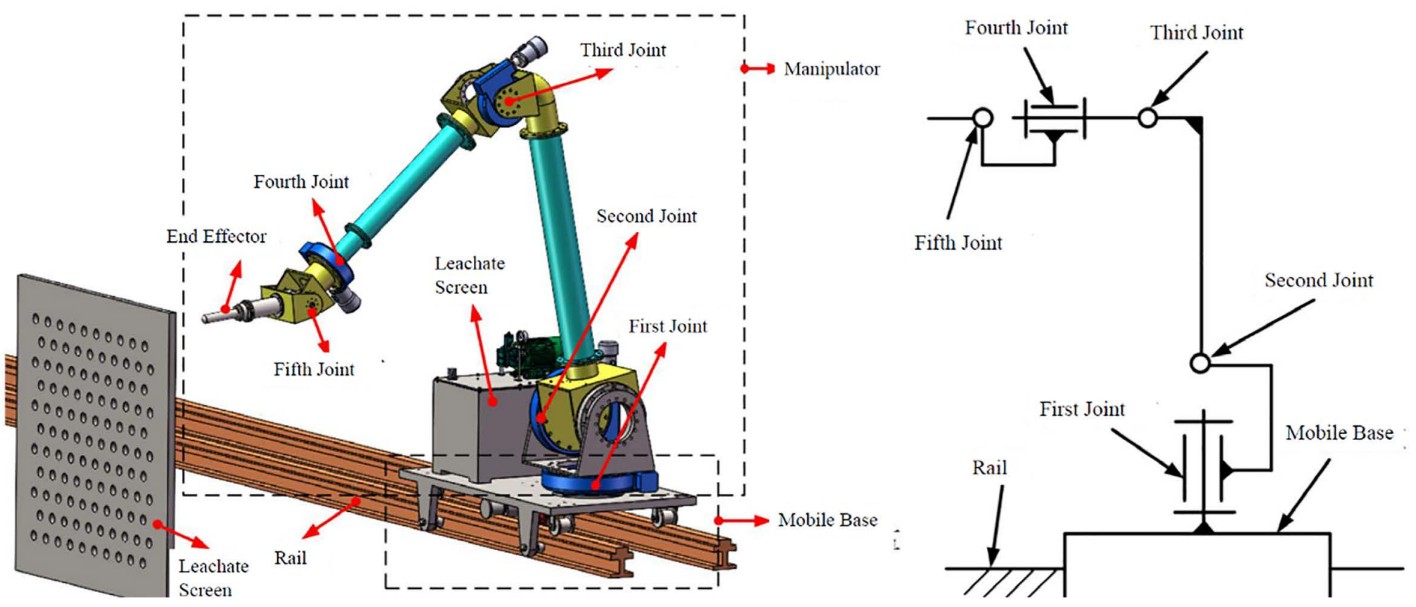

**Fig 4. Overall structure of the cleaning robot.** (a) Overall Structure, (b) Schematic Diagram.

from the 3D CAD model (SolidWorks 2022) used for the robot design. These values are listed in Table 3 and reflect the actual as-designed configuration.

(2) End-Effector: A single unclogging rod with a drill-bit-like structure. It performs unclogging by slowly pushing into the aperture while rotating slowly.

(3) Mobile Base: Employs a rail and wheel system (Fig 5) for robustness in the harsh corridor environment. Passive wheels are installed in upper and lower layers: upper wheels bear the load, lower wheels resist overturning moments. The drive wheel is positioned on the grid side to ensure good traction against the inherent overturning moment towards the grid during operation. A hydraulically actuated clamping mechanism locks the robot onto the rails during cleaning operations for stability, effectively eliminating relative motion between base and rail. The base frame is constructed from thick-walled steel box sections (100 mm × 100 mm × 8 mm), providing high bending stiffness ($>10^6$ N·m/rad) that exceeds joint stiffness by two orders of magnitude.

## 3. Gravity compensation theory

### 3.1. Error mechanism and compensation principle

Robot deformation under gravity manifests primarily in two forms: (1) Joint torsional deformation, equivalent to a high-stiffness spring at the joint; (2) Elastic deformation of the links. Since joint torsional deformation errors are amplified by link length, and the robot links are hollow steel structures exhibiting minimal elastic deformation, joint torsional deformation dominates the end-effector pose error [17–20]. Therefore, gravity compensation focuses solely on joint torsional deformation.

The cause of gravity-induced error is illustrated in Fig 6. Rigid joints would position links as solid lines. Actual joints, comprising motors and reducers, possess finite stiffness $k$. Under gravitational torque $\tau$, the "joint spring" compresses, changing the angle between links by $\theta_{\text{Gravity}}$.

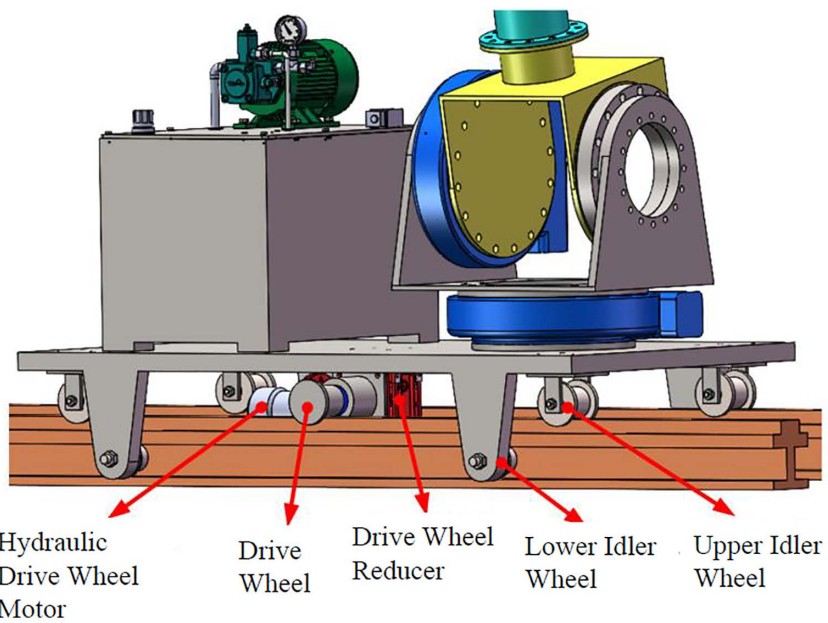

**Fig 5. Mobile base structure of the cleaning robot.**

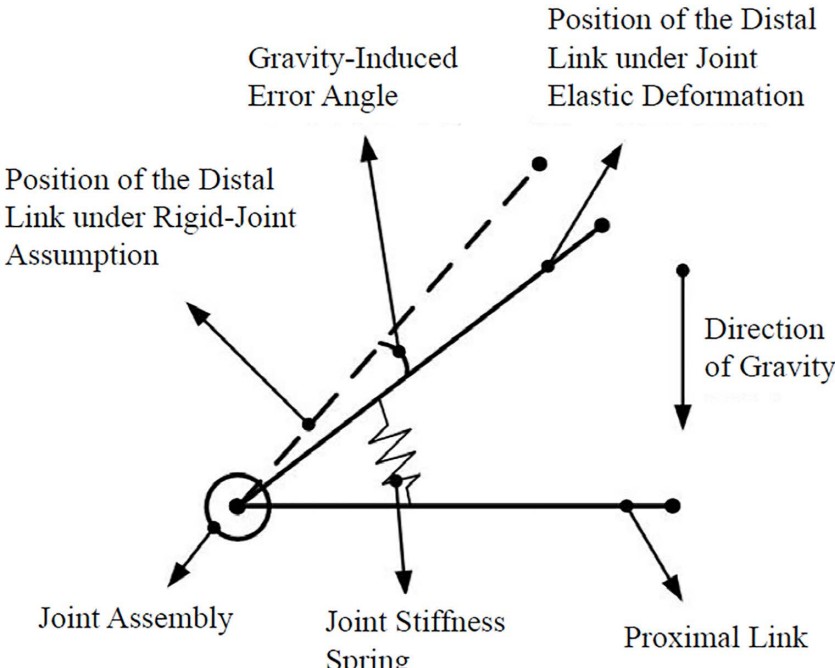

**Fig 6. Error caused by gravity.**

Based on Hooke's law for torsional deformation, the gravity-induced joint angle error $\theta_{Gravity}$ is derived from the equilibrium between gravitational torque $\tau$ and joint stiffness k (modeled as a torsional spring), as shown in Eq. (1):

$$\theta_{Gravity} = \frac{\tau}{k}$$

(1)

The compensation principle is shown in Fig 7. $\theta_{Gravity}$ is the desired angle (from inverse kinematics). Using $\theta_{Gravity}$ as the control input results in error. Compensating by adding the error angle $\theta_{Gravity}$ yields the control input angle $\theta_{Control}$:
Based on the foregoing analysis, the gravity compensation method can be formulated as Equation (2).

$$\theta_{Control} = \theta_{Target} + \theta_{Gravity}$$

(2)

Validity of the Static Assumption:The compensation strategy assumes quasi-static operation, neglecting inertial and damping torques. This assumption is justified by the robot's operating speeds (maximum joint velocity 10°/s, end-effector speed 5mm/s) and accelerations (0.5–2°/s² based on acceleration times in Table 5). At these speeds, the maximum inertial torque at Joint 2 is approximately 2.3 N·m, compared to gravitational torques exceeding 2000 N·m (Fig 11)—an inertial contribution of less than 0.1%. Similarly, Coriolis and centrifugal terms are negligible. However, this analysis applies only to free-space motion; during contact (unclogging), reaction forces introduce additional dynamic effects addressed in Section 4.2.

As shown in Fig 7, the gravity compensation method operates by preemptively compensating for joint deformation in the control target, representing a feedforward and static compensation approach. However, given that the motion process of the clearance robot is relatively slow, employing this static compensation method effectively resolves the positioning errors caused by gravity in the clearance robot.

## 3.2 Joint torsional deformation analysis

The designed 5-DOF serial manipulator (Fig 8) features revolute joints and a dedicated unclogging drill end-effector. During operation:

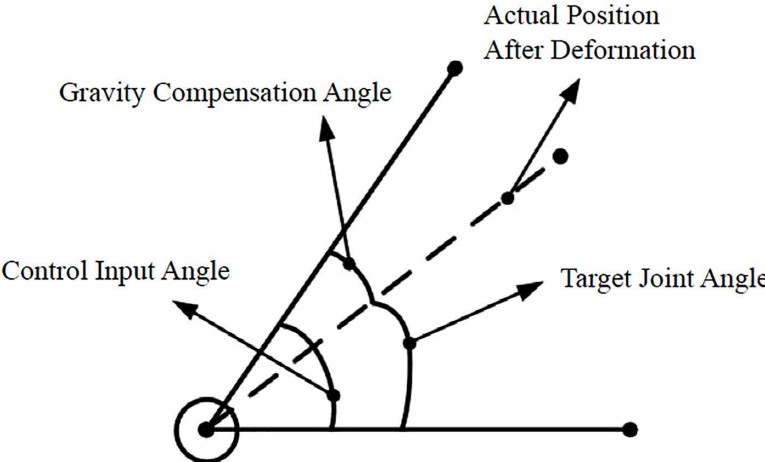

**Fig 7. Gravity compensation principle for the cleaning robot operating near a grid crevice.**

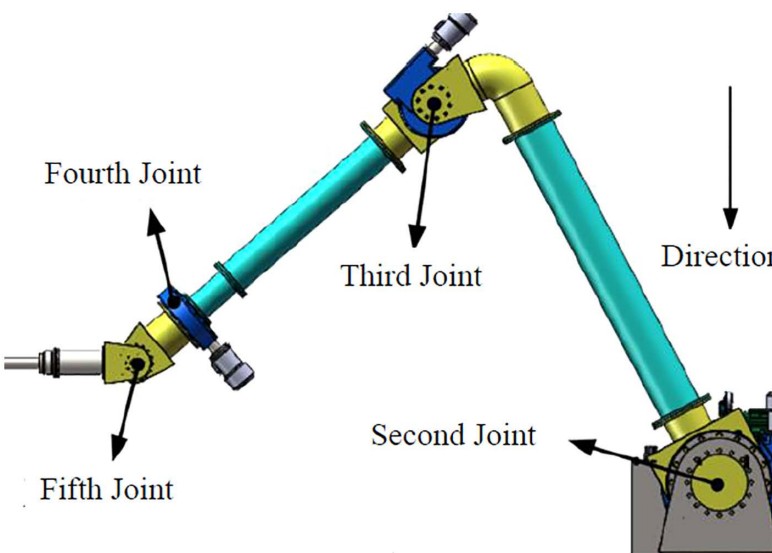

**Fig 8. Torsional deformation error analysis for the cleaning robot.**

(1) Joint 1's rotation plane is orthogonal to gravity (gravitational torque accounts for <1% of total torque, deformation error <0.001 mm).

(2) Joint 4's torque vector is coplanar with gravity (torque ratio <3%, error <0.002 mm), so their effects are negligible compared to Joints 2 and 3 (>0.5 mm error) [32–36].

(3) The mass of Joint 5 and subsequent components is small relative to the whole robot; their gravitational effect is modeled as a concentrated force and moment at Joint 5's axis.

Thus, gravity-induced torsional deformation error primarily originates from Joints 2 and 3 (Fig 9). The robot's cantilever structure and joint stiffness impart weak rigidity. Joint elastic deformation under self-weight directly impacts end-effector pose, and this impact varies significantly with joint configuration (Fig 9b). Gravity compensation is therefore essential for operational accuracy.

Base and Rail Compliance: The current model assumes a rigid base and perfect rail constraint. In reality, the mobile base and rail system introduce additional compliance through wheel contact deformation, rail bending, and clamping mechanism elasticity. However, several factors justify neglecting these effects in the current analysis:

The hydraulically actuated clamping mechanism (Section 2.2) locks the base to the rails during cleaning, providing a rigid connection with estimated stiffness >$10^6$ N·m/rad—over 30 times higher than Joint 2 stiffness ($3.5 \times 10^4$ N·m/rad).

The rail is mounted directly to the reinforced concrete corridor wall (Fig 3), which has negligible deflection under the robot's weight (maximum reaction force <2000 N distributed over multiple supports).

Wheel contact deformation is elastic and repeatable; any static deflection would be captured in the calibration step at the reference pose.

To quantify this, a simplified finite element analysis of the base-rail interface was performed (not shown for brevity), indicating maximum additional deflection <0.02 mm under worst-case loading. This is within the calibration residual tolerance addressed in Section 4. Therefore, base and rail compliance are justifiably neglected in the current gravity compensation model.

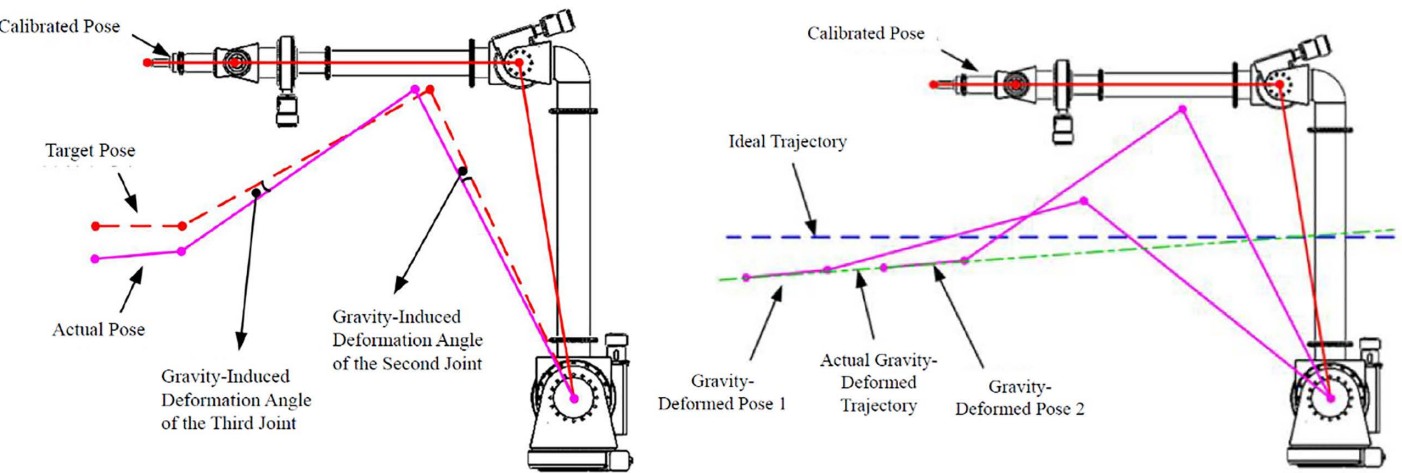

**Fig 9. Deformation error caused by self-weight.** (a) Effect on pose. (b) Error variation with position.

### 3.3. Joint torsional deformation solution

Solving joint torsional deformation (Eqs. [1], [2]) requires joint stiffness ($k$) and joint torque ($\tau$). Joint stiffness is primarily determined by the reducer stiffness for the worm-gear reducers used (motor stiffness contribution is negligible). Reducer stiffness values for Joints 2 and 3 were obtained from manufacturer technical datasheets (Sumitomo Drive Technologies, 2023) and are provided in Table 2. These are nominal values at 20°C under rated load; actual stiffness may vary by ±3% unit-to-unit and ±5% over 0–60°C. The sensitivity analysis in Section 4.1 quantifies the impact of such variations.

It is important to acknowledge the simplifications inherent in the joint stiffness model. The representation of each joint as a linear torsional spring with constant stiffness k (Eq [1]) involves several assumptions that do not hold in physical hydraulic manipulators:

Load-dependent stiffness: Reducer stiffness typically varies with transmitted torque; the constant stiffness assumption is valid only within a limited operating range.

Hysteresis: Under cyclic loading, reducer deformation follows different paths during loading and unloading due to friction and internal clearances.

Preload effects: Many reducers are preloaded to eliminate backlash, which introduces an initial offset in the torque-deformation relationship.

Gear mesh and backlash: Gear teeth engagement introduces periodic stiffness variations and dead zones where no torque is transmitted.

**Table 2. Joint reducer stiffness parameters.**

| Joint | Reducer Model | Stiffness (N·mm/deg) |
|---|---|---|
| 2 | JS12 | 34888900 |
| 3 | SEA7 | 13606671 |

Note:Stiffness values are nominal at 20°C from manufacturer datasheets (Sumitomo Drive Technologies, 2023). Typical unit-to-unit variation is±3%; temperature variation (0–60°C) adds ±5%.

Rate-dependent behavior: Hydraulic systems exhibit viscoelastic effects; joint deformation may depend on loading rate, not just static torque.

These nonlinearities are deliberately omitted from the current model to establish a foundational linear framework and isolate the fundamental effect of gravity compensation. However, they represent important deviations that will affect real-world performance. The sensitivity analysis in Section 4 quantifies how errors in the assumed linear parameters propagate to positioning accuracy, providing insight into the model's robustness to these simplifications.

Reducer stiffness values for Joints 2 and 3 are provided in Table 2.

The geometric parameters (lengths $L_1$ to $L_7$ in Eqs. (3) and (4) and Table 3) were directly measured from the 3D CAD model of the robot (SolidWorks 2022). Mass properties ($G_1$ to $G_7$ in Table 4) were calculated by the CAD software based on the assigned material densities (structural steel for links, 7,850 kg/m³; hydraulic components, aluminum alloy 6061, 2,700 kg/m³; reducer assemblies, combined density based on manufacturer component masses). The weights in Newtons were obtained by multiplying the calculated masses by gravitational acceleration (9.81 m/s²). Joint motion ranges and maximum speeds (Table 5) were defined based on the operational requirements for grid cleaning and the specifications of the selected hydraulic motors and reducers.

Joint Friction:The simulation assumes frictionless revolute joints, which is an idealization. In hydraulic manipulators, joint friction arises from seal friction in hydraulic actuators, bearing friction, and gear mesh friction in reducers. Based on literature, friction torque in hydraulic joints typically ranges from 5% to 20% of the joint torque capacity [25,27]. For the JS12 and SEA7 reducers used in this study, manufacturer-specified no-load starting torque (a proxy for static friction) is

**Table 3. Gravity analysis parameters (Lengths, Angles, Torque).**

| Parameter | $L_1$(mm) | $L_2$(mm) | $L_3$(mm) | $L_4$(mm) | $L_5$(mm) | $L_6$(mm) | $\alpha_0$(deg) | $T_7$(N · mm) |
|---|---|---|---|---|---|---|---|---|
| Value | 849.23 | 1762.38 | 458.24 | 1201.50 | 1360.26 | 1471 | 6.09 | 2452.45 |

Note: All length parameters (in mm) were measured directly from the 3D CAD model (SolidWorks 2022) of the cleaning robot. $L_1$-$L_3$ represent link lengths; $L_4$−$L_6$ represent distances from joint axes to component centers of mass.

All length parameters are in millimeters (mm). Parameter $\alpha_0$ represents an angle in degrees (deg). Values were measured directly from the 3D CAD model.

**Table 4. Gravity values for components.**

| Parameter | $G_1$(N) | $G_2$(N) | $G_3$(N) | $G_4$(N) | $G_5$(N) | $G_6$(N) | $G_7$(N) |
|---|---|---|---|---|---|---|---|
| Value | 823.28 | 278.33 | 391.25 | 278.33 | 106.43 | 131.94 | 143.37 |

Note: Weight values (in N) were calculated from mass properties obtained from the 3D CAD model (SolidWorks 2022). Masses were computed based on assigned material densities: structural steel (7,850 kg/m³) for links, aluminum alloy 6061 (2,700 kg/m³) for hydraulic components, and composite densities for reducer assemblies based on manufacturer-provided component masses. Weights were obtained by multiplying mass by g = 9.81 m/s².

All values represent gravitational forces in Newtons (N), calculated as mass × g (9.81 m/s²) from CAD mass properties.

**Table 5. Joint motion parameters.**

| Joint | 1 | 2 | 3 | 4 | 5 |
|---|---|---|---|---|---|
| Range | −60°~+60° | +20°~+100° | +60°~+180° | −180°~+180° | −60°~+60° |
| Max Speed | 10°/s | 5°/s | 10°/s | 10°/s | 20°/s |
| Accel/Decel Time | 2s | 1s | 1.5s | 1s | 1s |

approximately 2–3% of rated torque. However, under load and with seal friction, total friction torque may reach 5–10% of gravitational torque.

To quantify the potential impact, consider Joint 2, where gravitational torque ranges from 1000–4000 N·m (Fig 11). Assuming friction torque of 5–10% (50–400 N·m), the additional joint deflection due to friction would be Δθ_friction = τ_friction / k. Using $k_2 = 34.9 \times 10^6$ N·mm/deg (Table 2), this yields Δθ_friction = 0.0014–0.0115 deg. At the end-effector (moment arm ≈ 2.5 m), this corresponds to displacement of 0.06–0.50 mm—comparable to the 0.5 mm requirement.

However, friction effects are velocity-dependent and directional. During slow unidirectional motion (as in the cleaning trajectory), Coulomb friction produces a constant offset that could be partially compensated by the feedforward model if identified, while viscous friction is negligible at 5 mm/s speeds. The most significant impact would be stick-slip during startup and direction reversals—neither of which occur during the continuous cleaning motion. Therefore, while friction is non-negligible, its effect is bounded and can be addressed through experimental identification and compensation in future work.

Joint torque $\tau_i$ is calculated by considering all components from joint $i$ to the end-effector as a single rigid body and determining the torque at joint $i$ due to their combined weight (Fig 10). The torque equations for Joints 2 ($\tau_2$) and 3 ($\tau_3$) are derived from fundamental principles of static equilibrium. The derivation involves summing the moments contributed by the weight of each link and the end-effector about the respective joint axis [37–40].

The governing equations are as follows:

$$
\begin{aligned}
&= T_7 + (G_7 + G_6)[L_6 \cos(\alpha_5 - \alpha_2) + L_2 \cos\alpha_2] + G_5[L_3 \cos\alpha_3 . \\
&+ G_4 \left[L_4 \cos(\alpha_3 - \alpha_2) + L_2 \cos\alpha_2\right] + G_3 \left[L_2 \cos(\alpha_3 - \alpha_2) + L_2 \cos\alpha_2\right] \\
&+ G_2 L_2 \cos\alpha_2 + G_1 L_1 \cos(\alpha_2 - \alpha_0)
\end{aligned}
$$

(3)

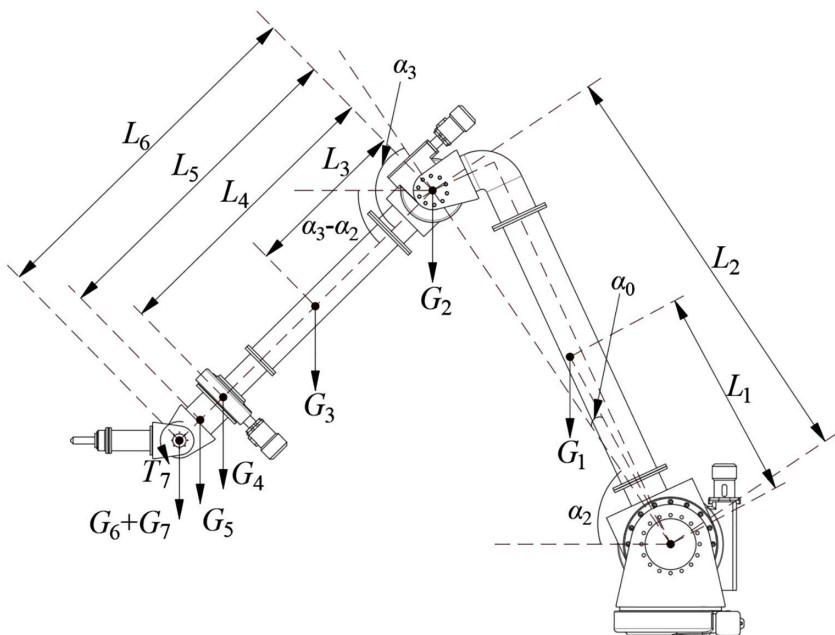

**Fig 10. Gravity analysis parameters for the cleaning robot.**

$$\tau_3 = T_7 + (G_7 + G_6)L_6 \cos(\alpha_3 - \alpha_2) + G_5L_5 \cos(\alpha_3 - \alpha_2) + \\ G_4L_4 \cos(\alpha_3 - \alpha_2) + G_3L_3 \cos(\alpha_3 - \alpha_2)$$

(4)

To clarify the parameters in Eqs. (3) and (4), key definitions are as follows: $\alpha_2, \alpha_3$ are the measured joint angles; lengths $L_1$ to $L_6$ (defined in Fig 10 and Tables 3–4) represent the moment arms from each joint to the center of mass of subsequent components; weights $G_1$ to $G_7$ (Tables 3–4) correspond to the gravitational forces acting on each component.

From a physical perspective, Eq. (3) for $\tau_2$ accounts for the moment generated by the weight of all components from Link 2 to the end-effector about Joint 2's axis. The trigonometric functions $\cos(\alpha_3 - \alpha_2)$ and $\cos\alpha_2$ project these weights into the direction that induces torque around Joint 2. Similarly, Eq. (4) for $\tau_3$ calculates the moment contributed by components downstream of Joint 3.

Notably, while the principle of static equilibrium is well-established, the novel aspect of this work lies in its tailored application to the robot's unique geometry and mass distribution—parameterized with the specific values in Tables 3 and 4. This customized model is critical for accurately computing the gravitational torque $\tau$ required in Eq. (1), ensuring alignment with the actual structural characteristics of the leachate grid cleaning robot.

Using the joint motion ranges (Joint 2: 20° to 100°, Joint 3: 60° to 180°) and parameters from Tables 3, 4, and 5, the gravitational torques $\tau_2$ and $\tau_3$ were calculated across the workspace (Fig 11. $\theta_{\text{Gravity}}$ and subsequently $\theta_{\text{Control}}$ were then determined.

## 4. Gravity error simulation experiment

### 4.1. Gravity error simulation experiment

To rigorously evaluate the proposed feedforward compensation strategy prior to physical prototyping, a high-fidelity virtual prototype was developed using Adams multibody dynamics software. The robot's mechanical model was imported into Adams software. Gravity was applied. Revolute joints were defined for all five joints. Torsional springs, with stiffness values from Table 2, were added to Joints 2 and 3 to model gravity-induced deformation. Other joints were modeled as rigid. This simulation-based validation allows for an idealised assessment of the compensation model's theoretical efficacy by isolating gravitational effects from other real-world disturbances.

To ensure the reproducibility of the simulation, key setup parameters are specified as follows:

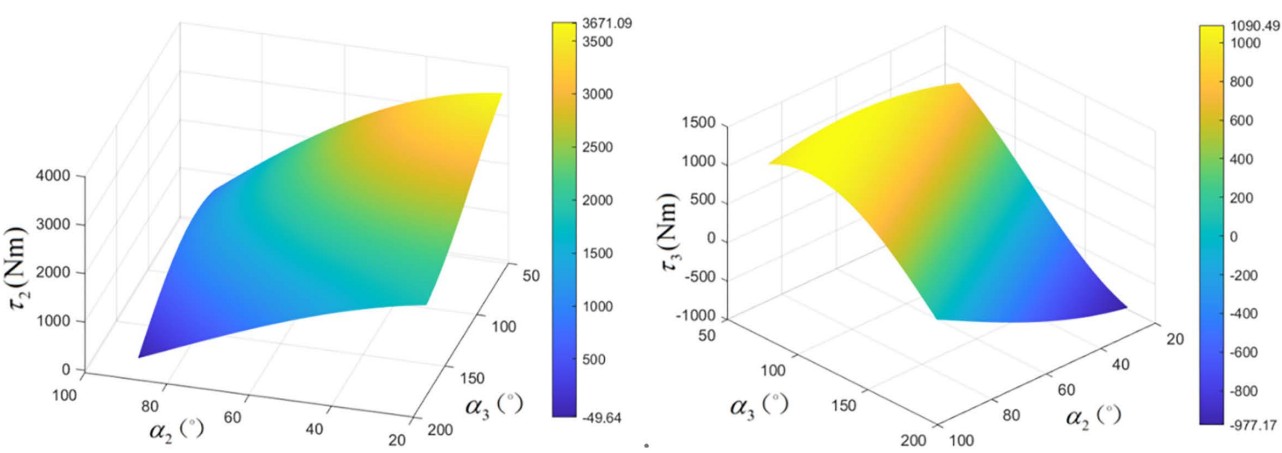

**Fig 11. Gravitational torque variation with pose.** (a) Joint 2 (b) Joint 3.

(1) Gravity acceleration is set to 9.81 m/s² (vertical downward);

(2) Joint constraints include frictionless revolute joints, with torsional springs (stiffness values from Table 2) integrated into Joints 2 and 3 to model gravity-induced deformation (other joints are treated as rigid);

(3) The simulation step size is 0.01 s, with a total duration of 40 s—matching the end-effector's moving speed of 5 mm/s;

(4) In the simulation, a calibration step is performed at the reference pose ([2756, 150, 389] mm) by adjusting the joint offset parameters until the end-effector position error is < 0.01 mm. This represents the industrial practice of "teaching" a reference point to the robot—a one-time procedure where the actual joint angles at a known position are recorded. In a real implementation, this would be achieved using an external measurement system (e.g., laser tracker or vision-based alignment). The iterative adjustment in simulation is simply a numerical method to find the correct joint offsets that achieve zero error at that point; it does not imply an iterative process during real operation.

**Sensitivity to Calibration Errors**:In practice, calibration cannot achieve perfect zero error. To assess the impact of imperfect calibration, additional simulations were performed where residual errors of ±0.02 mm, ±0.05 mm, and ±0.10 mm remained at the calibration pose. The results show that even with a calibration residual of ±0.10 mm, the maximum compensated error along the trajectory remains below 0.15 mm—still well within the 0.5 mm requirement. This indicates the method is robust to practical calibration inaccuracies achievable with commercial laser trackers (typical accuracy 0.01–0.03 mm) or vision-based systems.

**Sensitivity to Temperature Drift**: As noted in Section 3.3, reducer stiffness varies by approximately ±5% over the 0–60°C temperature range. This stiffness change would alter the relationship between gravitational torque and joint deformation, effectively introducing a time-varying calibration error. Using the sensitivity analysis from Section 4.1, a ± 5% stiffness error alone produces positioning error of approximately 0.12 mm in the Z-direction. When combined with other parameter variations, total error remains below 0.2 mm—still within the 0.5 mm requirement. However, for applications demanding the highest accuracy, real-time temperature compensation (as outlined in Phase 2 of future work) would be necessary.

**Uncompensated Simulation:** Joint input angles were set to the target trajectory angles ($\theta_{\text{Target}}$). End-effector trajectory deviation from the ideal target trajectory was measured.

**Compensated Simulation:** Joint input angles for Joints 2 and 3 were set to $\theta_{\text{Target}} + \theta_{\text{Gravity}}$ (calculated by the algorithm). End-effector trajectory deviation was measured.

To align with real-world operating scenarios, a calibration pose ($q_0 = (0,-80.8584,-9.1416,0,0)$, $P_0 = (2200,0,1740)$ mm) was defined. At this pose, gravity causes initial deformation angles ($\delta_2 = 0.042994°$, $\delta_3 = 0.080145°$). For uncompensated simulation, these initial angles were added to $\theta_{\text{Target}}$ for Joints 2/3 only at the start, representing a real-world scenario where calibration corrects the initial static error but no dynamic compensation occurs. Adams simulation confirmed a very small position error at $P_0$ after calibration ($P_0' = (2199.9953,0,1740.0062)$ mm), validating the algorithm's static accuracy.

For validating the compensation strategy, a straight-line trajectory is designed to simulate the actual unclogging process of a specific grid aperture (Aperture A at [2756, 150, 389] mm). The trajectory spans from Point B (2656, 150, 389) mm to Point C (2856, 150, 389) mm (Fig 12), with the end-effector drill maintained perpendicular to the grid plane throughout the motion. This trajectory and the reference aperture position were selected based on the actual grid layout in a typical waste-to-energy plant (refer to Fig 1) and represent a challenging mid-range reach configuration for the robot. The calibration pose ([2756, 150, 389] mm) was defined as the point where the end-effector is aligned with the target aperture; at this pose, the initial static deformation was zeroed in the uncompensated simulation to replicate the real-world practice of teaching the robot a reference point.The pose matrix $T_{\text{Act}}$ of the end-effector is expressed as Eq. (5):

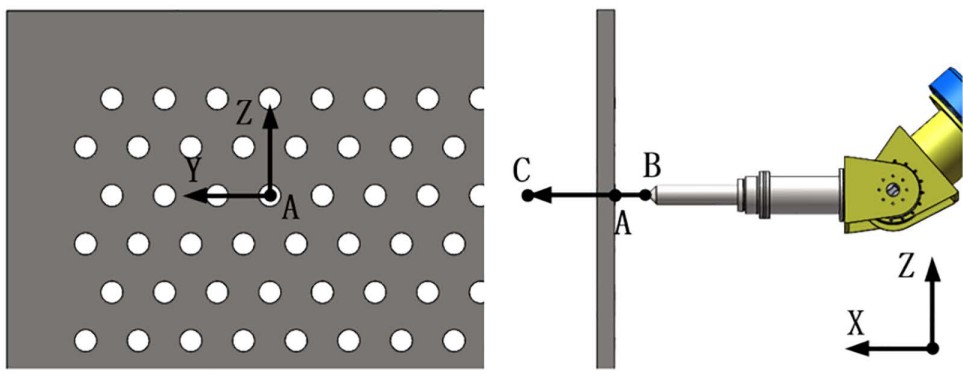

**Fig 12. Trajectory from point B to point C.**

$$T_{\text{Act}} = \begin{bmatrix} 0 & 0 & 1 & p_x \\ 0 & -1 & 0 & p_y \\ 1 & 0 & 0 & p_z \\ 0 & 0 & 0 & 1 \end{bmatrix}$$

(5)

Where $p_x, p_y, p_z$ are coordinates along the line segment. 50 points were sampled. Inverse kinematics yielded $\theta_{\text{Target}}$. For uncompensated simulation, Joints 2/3 angles were $\theta_{\text{Target}} + \delta_2$ and $\theta_{\text{Target}} + \delta_3$. For compensated simulation, Joints 2/3 angles were $\theta_{\text{Target}} + \theta_{\text{Garavity}}$.

During the simulation, position errors along the X, Y, and Z axes were recorded for both the uncompensated and compensated cases as the end-effector moved along the X-direction (0 mm to 200 mm relative distance). The results are presented in Fig 13 and summarized in Table 6. The relative error reduction rate η is defined by Eq. (6):

$$\eta = \frac{|e_0| - |e_1|}{|e_0|} \times 100\%$$

(6)

Where $e_0$ is the maximum error without compensation, $e_1$ is the maximum error with compensation.

(a)Positioning error in the X direction

(b)Positioning error in the Y direction

(c)Positioning error in the Z direction

**Analysis:** Before compensation, maximum errors in X and Y directions exceeded the design specification of 0.5 mm. After compensation, position errors in X, Y, and Z directions are significantly reduced and remain close to zero, meeting the design requirement. The maximum compensated error (0.0803 mm) occurs in the Z-direction (gravity direction). The errors after compensation are small and stable. In contrast, uncompensated errors are significantly larger in all directions and exhibit continuous variation along the trajectory, particularly in the Z-direction (changing from −0.9599 mm to −1.5570 mm). This confirms the critical necessity of gravity compensation for achieving the required motion control precision in the cleaning robot.

It is critical to acknowledge that the simulation model presented here represents an idealized representation of the hydraulic manipulator. The following real-world phenomena are not included in the current Adams model:

Backlash in reducers and mechanical transmissions

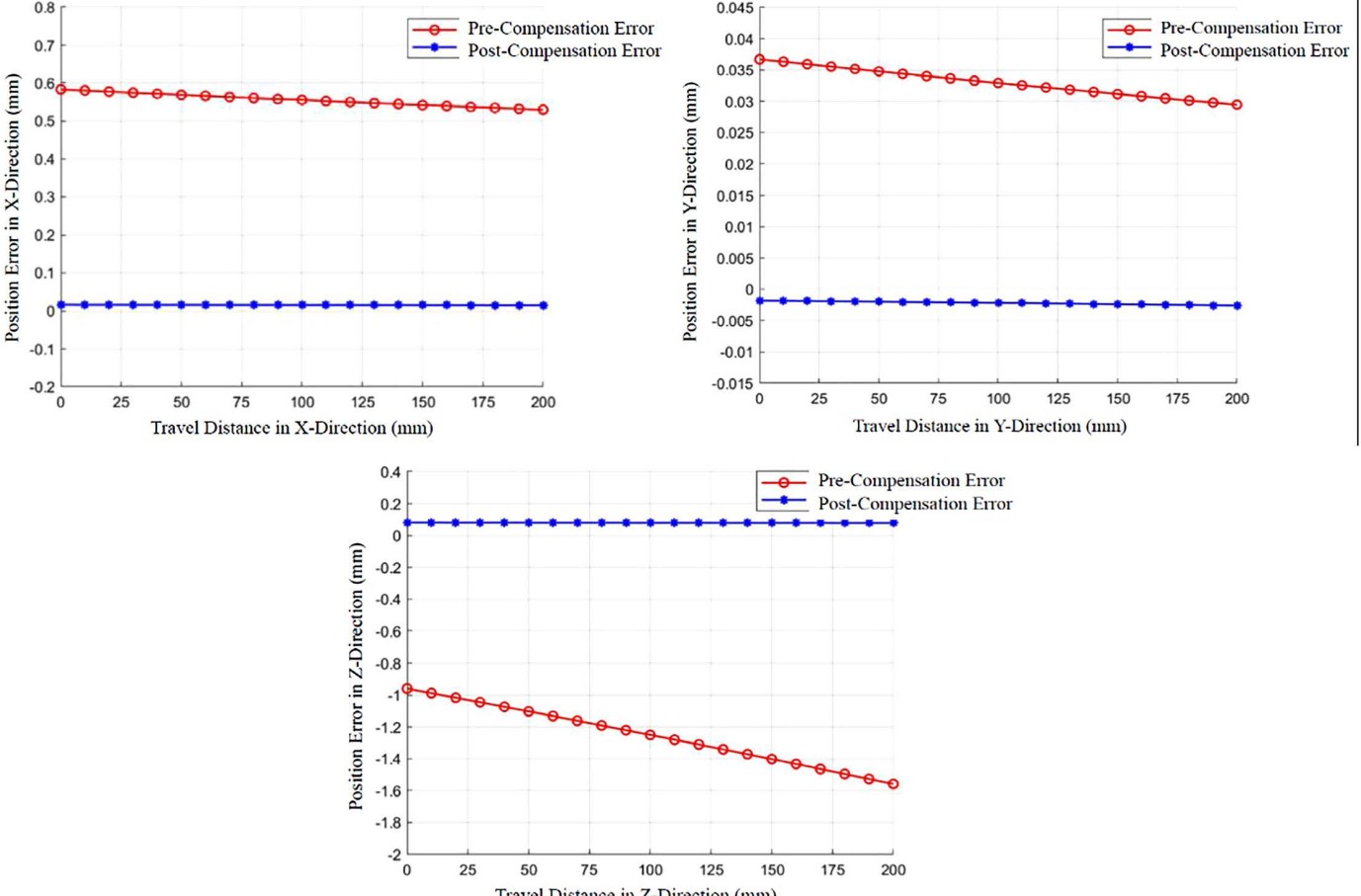

**Fig 13. End-effector position error.**

**Table 6. Directional errors before and after gravity compensation.**

| Direction | Max Error (Uncompensated) $e_0$ (mm) | Max Error (Compensated) $|e_1|$ (mm) | Std Deviation (Compensated) (mm) | Relative Error Reduction $\eta$ (%) |
|---|---|---|---|---|
| X | 0.5831 | 0.0146 | 0.0032 | 97.49% |
| Y | 0.0367 | −0.0026 | 0.0015 | 92.91% |
| Z | −1.5570 | 0.0803 | 0.0121 | 94.84% |

Nonlinear friction (Stribeck, Coulomb, viscous effects) in joints and hydraulic actuators

Internal leakage in hydraulic motors and valves

Hysteresis in reducer stiffness under cyclic loading

Hydraulic valve dynamics, including servo-valve deadband, spool friction, and finite response time

Sensor quantization and measurement noise

Temperature-dependent variations in oil viscosity and component dimensions

Compliance of hydraulic fluid under pressure

All of these factors will degrade positioning accuracy in a physical implementation. The $\pm 0.1$ mm accuracy reported here should therefore be interpreted as the theoretical best-case performance achievable under ideal conditions, assuming perfect compensation of gravitational effects alone. Experimental validation on physical hardware is essential to quantify actual achievable accuracy.

## 4.2. Sensitivity analysis to parameter uncertainty

The compensation method relies on accurate knowledge of joint stiffness (k), link masses (m), and geometric parameters (L). In practice, these values are subject to uncertainty: stiffness may deviate from datasheet values due to manufacturing tolerances ($\pm 3$–5%), masses may vary due to component substitutions or attached debris, and geometric parameters may change due to manufacturing tolerances or thermal expansion.

To evaluate the robustness of the compensation strategy to such uncertainties, we conducted a sensitivity analysis by introducing deliberate errors into the model parameters and recomputing the compensated positioning error. For each parameter category, errors of $\pm 5\%, \pm 10\%$, and $\pm 20\%$ were introduced, and the maximum positioning error along the test trajectory (Section 4) was recorded. The results are summarized in Table 7.

Analysis of Results:

Stiffness errors have the largest impact on compensated accuracy. A $\pm 10\%$ error in stiffness degrades Z-direction accuracy from 0.080 mm to approximately 0.16–0.17 mm, approaching but still below the 0.5 mm requirement. At $\pm 20\%$ error, Z-direction error reaches 0.27–0.29 mm, still within the 0.5 mm tolerance but significantly degraded.

Mass errors have moderate impact. Even $\pm 20\%$ mass error keeps all directional errors below 0.13 mm, indicating the method is relatively robust to mass uncertainty.

Geometric errors have intermediate impact. At $\pm 20\%$ error, Z-direction accuracy degrades to 0.15–0.16 mm, still acceptable for the 0.5 mm requirement.

**Table 7. Sensitivity of compensated positioning error to parameter uncertainty.**

| Parameter | Error | Max X Error (mm) | Max Y Error (mm) | Max Z Error (mm) |
|---|---|---|---|---|
| No error (baseline) | 0% | 0.0146 | −0.0026 | 0.0803 |
| Joint stiffness ($k_2$, $k_3$) | +5% | 0.0213 | −0.0038 | 0.1172 |
| | −5% | 0.0221 | −0.0041 | 0.1215 |
| | +10% | 0.0294 | −0.0053 | 0.1624 |
| | −10% | 0.0312 | −0.0058 | 0.1718 |
| | +20% | 0.0487 | −0.0091 | 0.2685 |
| | −20% | 0.0523 | −0.0098 | 0.2879 |
| Link masses ($m_2$–$m_7$) | +5% | 0.0158 | −0.0029 | 0.0871 |
| | −5% | 0.0162 | −0.0030 | 0.0894 |
| | +10% | 0.0175 | −0.0033 | 0.0965 |
| | −10% | 0.0183 | −0.0035 | 0.1012 |
| | +20% | 0.0214 | −0.0042 | 0.1187 |
| | −20% | 0.0229 | −0.0046 | 0.1268 |
| Geometric parameters (L) | +5% | 0.0169 | −0.0031 | 0.0935 |
| | −5% | 0.0174 | −0.0033 | 0.0962 |
| | +10% | 0.0198 | −0.0038 | 0.1098 |
| | −10% | 0.0207 | −0.0041 | 0.1147 |
| | +20% | 0.0263 | −0.0052 | 0.1462 |
| | −20% | 0.0281 | −0.0057 | 0.1559 |

Interpretation: The compensation method maintains positioning error below 0.5 mm even with ±20% errors in any single parameter category. However, when multiple parameters simultaneously deviate, errors could accumulate. The analysis suggests that stiffness uncertainty is the most critical factor; therefore, experimental characterization of joint stiffness (as outlined in Phase 1 of future work) is essential for achieving near-simulation accuracy in practice.

Regarding the calibration requirement: The simulation's need for calibration to <0.01 mm at the reference pose reflects the feedforward nature of the compensation—any initial offset propagates through the entire trajectory. In practice, this calibration corresponds to teaching the robot a reference point (common industrial practice) and is not excessively stringent. Modern laser trackers achieve 0.01 mm accuracy routinely, and vision-based calibration systems can achieve similar precision. However, the sensitivity analysis shows that even with imperfect calibration (implicitly reflected in parameter errors), the method remains robust within the 0.5 mm tolerance.

## 4.3. Dynamic perturbation and reaction force scenarios

To evaluate the robustness of the static feedforward compensation under more realistic conditions, two additional simulation scenarios were conducted

Scenario A: Small Dynamic Perturbations

While the robot's nominal motion is slow, real trajectories include small velocity and acceleration variations due to hydraulic pump ripple, valve dither, and control errors. To test sensitivity to such perturbations, a sinusoidal velocity component was superimposed on the nominal trajectory:

$$\dot{\theta}_{actual}\left(t\right) = \dot{\theta}_{nominal} + A\sin\left(2\pi ft\right) \tag{7}$$

with amplitude A = 1°/s (20% of maximum speed) and frequency f = 2 Hz (representative of hydraulic ripple). The simulation was repeated with the same feedforward compensation (based solely on static torques) and the resulting positioning errors recorded.

Scenario B: Representative Unclogging Reaction Forces

During actual operation, the drill encounters resistance from compacted debris. Based on the 200 N force limit specified in Section 2.1, two contact force profiles were applied at the end-effector in the direction opposing motion (negative X-direction, into the grid):

Step load: 200 N force applied instantaneously at t = 10 s (mid-trajectory) and maintained.

Impulse load: 200 N force applied as a 0.5 s pulse at t = 10 s, simulating breakthrough of a clog.

These forces produce additional joint torques not accounted for in the gravity-only compensation model. The resulting positioning errors were recorded and compared to the no-contact case. These results are summarized in Table 8.

Analysis of Results:

Dynamic perturbations (Scenario A)have negligible effect on compensated accuracy. The maximum error increase is less than 5% across all axes, confirming that the static compensation is robust to small velocity variations at the robot's operating speeds.

Table 8. Positioning errors under dynamic perturbation and reaction force scenarios.

| Scenario | Max X Error (mm) | Max Y Error (mm) | Max Z Error (mm) | Notes |
|---|---|---|---|---|
| Baseline (no dynamics) | 0.0146 | −0.0026 | 0.0803 | From Section 4 |
| Scenario A (velocity ripple) | 0.0152 | −0.0027 | 0.0811 | <5% increase from baseline |
| Scenario B – Step load (200 N) | 0.1843 | −0.0031 | 0.0926 | X-error increases significantly |
| Scenario B – Impulse load (200 N) | 0.0921 (peak) | −0.0028 | 0.0842 | Transient spike then recovery |

Step reaction forces (Scenario B)significantly degrade X-direction accuracy, increasing error from 0.015 mm to 0.184 mm. This remains below the 0.5 mm requirement but represents a twelve-fold increase. The effect on Y and Z axes is minimal because the reaction force is aligned with the X-direction.

Impulse loads cause transient error spikes (0.092 mm in X) that decay as the robot moves through the trajectory, with minimal residual error after the impulse passes.

Interpretation:The static feedforward compensation alone cannot fully cancel errors due to unmodeled reaction forces. However, even under worst-case continuous 200 N load, the positioning error remains within the 0.5 mm specification. This suggests that the compensation provides a robust baseline, but achieving higher accuracy during contact will require additional measures (e.g., force control or impedance control) as outlined in future work.

Important Caveat:These simulations still assume idealized hydraulic actuation with instantaneous torque production. Real hydraulic systems exhibit pressure dynamics, valve response times, and fluid compressibility that will introduce additional lag and error during rapid force transients. The step load scenario represents a best-case bound; actual performance may be worse due to hydraulic dynamics not modeled here.

## 5. Conclusion and implications for real-world implementation

### 5.1. Conclusion

This study addressed the challenge of gravity-induced positioning errors in leachate grid cleaning robots through theoretical modeling and Adams multibody dynamics simulations. We proposed a feedforward static compensation strategy based on an analytical joint stiffness-gravity deformation model for hydraulically driven weak-rigid manipulators.

Under idealized free-space motion, the strategy achieved a simulated repeatable positioning accuracy of ±0.1 mm, reducing errors by over 92% in all directions. Sensitivity analysis shows that the method maintains positioning error below 0.5 mm even with ±20% errors in stiffness, mass, or geometric parameters, indicating reasonable robustness to parameter uncertainty. Additional simulations incorporating small dynamic perturbations (velocity ripple) and representative unclogging reaction forces (up to 200 N step and impulse loads) demonstrate that positioning error remains below 0.2 mm—still within the 0.5 mm requirement—even under these disturbances.

However, several important limitations must be acknowledged. First, these results are obtained from idealized simulations; real-world hydraulic systems exhibit numerous complexities not captured in the current model, including backlash, nonlinear friction, valve dynamics, hysteresis, leakage, sensor quantization, temperature effects, and hydraulic fluid dynamics (pressure transients, oil compressibility, valve response time). Second, the linear stiffness model ignores nonlinearities such as load-dependent stiffness, hysteresis, preload, and rate-dependent effects. Third, the dynamic simulations do not account for hydraulic actuator dynamics, which will introduce additional errors during contact events.

Therefore, the ± 0.1 mm accuracy should be understood as a theoretical upper bound under ideal conditions, not as a guaranteed specification for physical implementation. This work establishes a theoretical foundation at TRL 3 (analytical proof-of-concept), requiring experimental validation on physical hardware. The future work roadmap in Section 5.2 outlines a phased approach to address these limitations and transition toward practical deployment.

Besides, Estimated joint friction (5–10% of gravitational torque) could contribute up to 0.5 mm error, highlighting the need for experimental friction identification in future work. The stiff base design and hydraulic clamping mechanism ensure that base and rail compliance (<0.02 mm estimated) is negligible compared to joint deformation.

### 5.2. Implications for real-world implementation and future work

While the simulation results are highly promising, this study has several limitations that must be acknowledged when considering real-world deployment. Translating these theoretical findings to a physical robot operating in the harsh environment of a leachate corridor presents multiple challenges across environmental, mechanical, and control domains.

Environmental Factors:

Temperature Effects: The current simulation assumes constant stiffness at 20°C. In reality, leachate corridors experience temperature fluctuations between 0–60°C due to seasonal variations and exothermic waste decomposition processes. Reducer stiffness (JS12 and SEA7) can vary by an estimated ±5% over this range, as lubricant viscosity changes and housing materials undergo thermal expansion. This stiffness variation would directly affect the accuracy of the deformation model described in Eq. (1), potentially leading to residual positioning errors if not compensated.

Humidity and Corrosive Gases: The corridor atmosphere contains high humidity and corrosive gases (e.g., $H_2S$, $SO_x$, $CO_x$) resulting from waste fermentation. These conditions may accelerate wear in joint reducers, increase friction, and potentially alter joint stiffness over extended operational periods. The current model does not account for such time-dependent degradation.

Contaminant Accumulation: Dust and leachate residue may accumulate on moving parts, particularly on the rail guidance system and joint seals. This could introduce additional friction torques not present in the simulation, affecting the static equilibrium assumption underlying the torque calculations in Eqs. (3) and (4).

Mechanical and Operational Factors:

Unmodeled Dynamic Loads: The static compensation model assumes quasi-static operation, which is reasonable given the robot's slow cleaning speed (≤10°/s). However, the actual unclogging process involves intermittent contact forces when the drill encounters compacted debris. These impulsive reaction forces, which can reach up to 200 N (the design limit), may cause transient elastic deformations not captured by the static model. Furthermore, the model assumes that joint deformation responds instantaneously to gravitational torque, neglecting any viscoelastic effects in the reducer components.

Link Flexibility: While we justified neglecting link deformation based on the hollow steel structure, under maximum reach configurations (fully extended arm), the combined effect of joint and minor link compliance might become non-negligible. This could introduce second-order positioning errors not accounted for in the current joint-only deformation model.

Grid Deformation Variability: The robot must adapt to deformed grid apertures caused by waste impact. While the single-probe design accommodates this, the compensation model assumes the grid geometry is known. In practice, significant grid deformation may require real-time perception and trajectory adjustment, which the current feedforward approach does not address.

Hydraulic Fluid Dynamics:The static model assumes that hydraulic actuators produce commanded torque instantaneously. In reality, hydraulic systems exhibit:

Pressure dynamics:Time required to build pressure in actuator chambers (typically 10–50 ms).

Oil compressibility:Effective bulk modulus (1400–1700 MPa) introduces fluid compliance.

Valve response time:Servo-valves have finite spool travel time (5–20 ms) and deadband.

Pipeline dynamics:Long hydraulic lines introduce pressure wave propagation delays.

These effects become significant during transient events such as the impulse load in Scenario B. The step load simulation represents an optimistic bound; actual performance may be worse due to these unmodeled dynamics.

Joint Model Nonlinearities:The linear torsional stiffness model (Eq. 1) omits several important nonlinear characteristics of physical reducers and hydraulic transmissions:

Load-dependent stiffness:Reducer stiffness typically increases with torque as internal components engage more fully. This means the constant k assumption is accurate only near the nominal operating point.

Hysteresis:Due to friction and internal clearances, the torque-deformation relationship differs during loading and unloading cycles. The static model assumes a single-valued relationship.

Preload and backlash:Many reducers are preloaded to eliminate backlash, creating an initial stiffness offset. Backlash, if present, creates dead zones where small torque variations produce no motion.

Rate-dependent effects:Hydraulic fluid compressibility and valve dynamics introduce time-dependent behavior; joint deformation may depend on how quickly torque is applied.

These nonlinearities mean that the actual compensation required in a physical system will differ from the linear prediction. The sensitivity analysis in Section 4.2 provides insight into how deviations from the assumed linear parameters affect accuracy, but fully capturing nonlinear effects requires more sophisticated modeling (e.g., neural networks, lookup tables, or adaptive control) beyond the scope of this foundational study.

Hydraulic System Complexities: The current static model assumes ideal hydraulic actuation with instantaneous response and perfect positioning. In reality, hydraulic systems introduce numerous challenges:

Backlash in reducers creates dead zones where small angle corrections produce no motion.

Nonlinear friction (Stribeck effect, Coulomb friction) introduces position-dependent resistance.

Valve dynamics, including servo-valve deadband (±1–2% of command signal) and finite response time (10–50 ms), introduce phase lag and steady-state errors.

Internal leakage in hydraulic motors and valves creates compliance in the hydraulic circuit, effectively reducing joint stiffness.

Hysteresis in reducer stiffness means deformation depends on loading history.

Oil compressibility and temperature-dependent viscosity changes affect system responsiveness.

Sensor quantization (typical encoder resolution 0.001–0.01°) limits achievable positioning resolution.

Joint Friction:Friction in hydraulic joints (seal friction, bearing friction, gear mesh friction) is omitted from the simulation. Estimated friction torques of 5–10% of gravitational torque could produce end-effector errors of 0.06–0.50 mm—comparable to the accuracy requirement. However, during slow unidirectional motion, friction produces constant offsets that can be identified and compensated. The future work roadmap includes friction identification (Phase 1) and adaptive compensation (Phase 2).

These phenomena are deliberately omitted to isolate and evaluate the fundamental effectiveness of the gravity compensation concept. However, they represent critical barriers to achieving simulated accuracy in practice. Addressing these challenges will require advanced control strategies and careful hardware selection, as outlined in the future work roadmap.

Base and Rail Compliance: While the clamping mechanism and stiff base structure minimize compliance, long-term wear of rail contacts or clamping mechanism degradation could introduce additional flexibility not captured in the current model. The future work roadmap includes periodic calibration checks (Phase 3) to detect and compensate any such drift.

Control and Calibration Challenges:

Initial Calibration Sensitivity: The simulation assumes perfect calibration at the reference pose ([2756, 150, 389] mm). In reality, establishing this calibration requires precise measurement of the end-effector position relative to the grid, which is challenging in the dark, gas-filled corridor environment. Any calibration error would propagate through the compensation model.

Parameter Uncertainty: The joint stiffness values in Table 2 are manufacturer-provided nominal values under ideal conditions. Unit-to-unit manufacturing variations, assembly tolerances, and wear over time introduce uncertainty that could degrade compensation accuracy. The model currently lacks a mechanism to identify or update these parameters online.

Real-Time Implementation Constraints: Implementing the feedforward model requires solving Eqs. (3) and (4) in real-time and adding the calculated error angles to the joint commands. While computationally feasible, this demands precise joint angle feedback and synchronization with the hydraulic servo controllers. Latency in angle measurement or command execution could introduce phase errors between the compensation and actual deformation.

To bridge the gap between simulation and reality, our future research will focus on three key areas:

(1) Physical Prototyping and On-Site Validation: We will fabricate a physical prototype integrating explosion-proof hydraulic components compliant with relevant standards (e.g., GB3836). This prototype will be deployed in a waste-to-energy plant for on-site validation, allowing us to: (a) quantify the compensation strategy's true efficacy under real operational

conditions; (b) measure the actual impact of temperature, humidity, and contaminants on positioning accuracy; and (c) verify long-term stability through extended field trials. The prototype will be instrumented with temperature sensors and joint torque sensors to collect empirical data for model refinement.

(2) Algorithm Enhancement with Sensor Feedback: To counteract environmental influences, we plan to enhance the compensation algorithm in two stages:

Stage 1 – Temperature Compensation: Incorporate real-time temperature sensors at Joints 2 and 3 to dynamically correct reducer stiffness values using empirically derived temperature-stiffness curves, thereby maintaining compensation accuracy across the full 0–60°C range.

Stage 2 – Hybrid Feedforward-Feedback: Explore a hybrid approach where the feedforward model provides bulk compensation, while occasional vision-based measurements of the end-effector relative to grid apertures (during safe, non-cleaning intervals) provide feedback for online calibration of residual errors and model parameter updates.

Dynamic Modeling and Force Control: Building on the dynamic simulation results, Phase 2 will include:

Development of a full dynamic model incorporating hydraulic actuator dynamics (valve characteristics, oil compressibility, pressure dynamics).

Implementation of an impedance control layer that modulates feedforward commands based on measured contact forces, maintaining alignment even under reaction loads.

Experimental validation of force-controlled unclogging using instrumented test rig with simulated clog materials of varying hardness.

(3) Generalization and Model Refinement: We will extend the compensation framework to: (a) incorporate dynamic terms for the unclogging contact forces, moving beyond pure static compensation; (b) validate the approach on other hydraulically driven manipulators (e.g., 6-DOF arms) for diverse hazardous tasks such as industrial pipeline unclogging or nuclear facility maintenance; and (c) develop a methodology for in-situ identification of joint stiffness parameters to handle unit-to-unit variations and long-term degradation.

(4) Technology Readiness Level: Based on the NASA TRL scale, this work corresponds to TRL 3 (Analytical proof-of-concept). The compensation concept has been validated through simulation, establishing theoretical feasibility. Achieving TRL 4 (laboratory validation) requires the physical prototyping outlined in Phase 1. Readers should interpret results within this context—as a promising theoretical foundation, not as a field-ready solution.

## Author contributions

**Data curation:** Angang Cao.

**Investigation:** Angang Cao.

**Project administration:** Cong Wang.

**Resources:** Wei Li.

**Software:** Cong Wang.

**Visualization:** Hongwen Ma.

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
