## [Decision Letter · Decision Letter 0]

1 Feb 2026

PONE-D-25-68774Gravity Compensation for Leachate Grid Cleaning Robots in Waste-to-Energy Plants: A Modeling and Simulation StudyPLOS One

Dear Dr. Cao,

Thank you for submitting your manuscript to PLOS ONE. After careful consideration, we feel that it has merit but does not fully meet PLOS ONE’s publication criteria as it currently stands. Therefore, we invite you to submit a revised version of the manuscript that addresses the points raised during the review process.

A letter that responds to each point raised by the academic editor and reviewer(s). You should upload this letter as a separate file labeled ‘Response to Reviewers’.A marked-up copy of your manuscript that highlights changes made to the original version. You should upload this as a separate file labeled ‘Revised Manuscript with Track Changes’.An unmarked version of your revised paper without tracked changes. You should upload this as a separate file labeled ‘Manuscript’.

We look forward to receiving your revised manuscript.

Kind regards,

Shamshad Alam, PhD

Academic Editor

PLOS One

Journal Requirements:

1. Please ensure that your manuscript meets PLOS ONE’s style requirements, including those for file naming. The PLOS ONE style templates can be found at

“This work was supported by the Henan Province Science and Technology Research Project(No.252102220033),  Natural Science Foundation of Henan（No.252300420072） and  the education department of Henan Province (No. 24A460023 and No. 26A460029).”

“This work was supported by the Henan Province Science and Technology Research Project(No.252102220033),  Natural Science Foundation of Henan（No.252300420072） and  the education department of Henan Province (No. 24A460023 and No. 26A460029).”

“This work was supported by the Henan Province Science and Technology Research Project(No.252102220033),  Natural Science Foundation of Henan（No.252300420072） and  the education department of Henan Province (No. 24A460023 and No. 26A460029).”

“This work was supported by the Henan Province Science and Technology Research Project(No.252102220033),  Natural Science Foundation of Henan（No.252300420072） and  the education department of Henan Province (No. 24A460023 and No. 26A460029).”

6. Thank you for stating the following in your Competing Interests section:

“The authors declare that they have no known competing financial interests or personal relationships that could have appeared to influence the work reported in this paper.”

7. In the online submission form, you indicated that the data supporting this study’s findings are available from the corresponding author upon reasonable request.

8. We note that Figures 1, 4, 5 and 8 in your submission contain copyrighted images. All PLOS content is published under the Creative Commons Attribution License (CC BY 4.0), which means that the manuscript, images, and Supporting Information files will be freely available online, and any third party is permitted to access, download, copy, distribute, and use these materials in any way, even commercially, with proper attribution. For more information, see our copyright guidelines: http://journals.plos.org/plosone/s/licenses-and-copyright.

a. You may seek permission from the original copyright holder of Figures 1, 4, 5 and 8 to publish the content specifically under the CC BY 4.0 license.

Reviewers' comments:

Reviewer's Responses to Questions

**Comments to the Author**

1. Is the manuscript technically sound, and do the data support the conclusions?

Reviewer #1: Partly

Reviewer #2: Yes

2. Has the statistical analysis been performed appropriately and rigorously? 

Reviewer #1: Yes

Reviewer #2: Yes

3. Have the authors made all data underlying the findings in their manuscript fully available?

Reviewer #1: Yes

Reviewer #2: Yes

4. Is the manuscript presented in an intelligible fashion and written in standard English?

Reviewer #1: Yes

Reviewer #2: Yes

5. Review Comments to the Author

Reviewer #1: This manuscript addresses a relevant engineering problem with a technically sound methodology. While the work is limited to simulation, it provides a solid theoretical foundation for future physical implementation. The writing is clear and the results are promising. I recommend minor revision to address the concerns outlined in the following, particularly regarding validation limitations and parameter documentation.

1. Expand validation discussion:Clarify that results are simulation-only and discuss implications for real-world implementation.

2. Strengthen limitations section:Include discussion of factors that could affect real-world performance.

3. Clarify parameter sources: Provide more detail on how stiffness values and other parameters were determined.

4. Technical corrections:

- Remove duplicate phrase at line 334

- Verify units consistency in Tables 3 and 4

- Improve figure placement relative to in-text references

5. Expand future work: Given that this is simulation-only research, the future work section should more clearly outline the path to physical implementation.

Reviewer #2: The manuscript presents a feedforward static gravity-compensation method for a hydraulically driven, rail-mounted 5-DOF manipulator intended to clean clogged leachate grids in waste-to-energy plants. The authors derive closed-form expressions for gravity-induced joint torques for Joints 2 and 3, model torsional compliance using reducer stiffness values (JS12 and SEA7), compute gravity-induced joint deflections via Hooke’s law, and apply a pre-compensation angle to the control inputs. The approach is validated entirely in Adams multibody simulations using the authors’ robot geometry and mass parameters; reported results show dramatic reductions in positioning error (≈92–97% reduction), with residual errors within ±0.1 mm, meeting the stated 0.5 mm engineering requirement. The paper is interesting and it can be published if the authors address the following concerns:

The conclusions repeatedly imply readiness for field deployment but all results are from idealized Adams simulations. Real hydraulic systems exhibit backlash, leakage, nonlinear friction, hysteresis in reducers, valve dynamics, servo-valve deadband, and sensor quantization—none of which are modeled. The claim of ±0.1 mm repeatable accuracy in a hydraulic manipulator (long-reach, weak-rigid) without closed-loop correction is unrealistic unless proven on hardware. The limitation is acknowledged (briefly) but is not treated with sufficient caution in the conclusions. Provide explicit caveats and reduce overstated claims until experimental data exist.

Modeling reducer + transmission as a single linear torsional stiffness (k) ignores nonlinearities (stiffness varying with angle/load), hysteresis and preload, and gear mesh/backlash. For hydraulic drives, compliant behavior may include rate-dependent terms. The simulation forces precise calibration (calibration to <0.01 mm) to succeed—this suggests the method is brittle to parameter uncertainty. Authors must quantify sensitivity of compensation to ±X% errors in stiffness, mass, and geometry (sensitivity analysis).

The method is static feedforward; yet the robot moves (albeit slowly). The paper assumes motions are slow enough to ignore dynamic torques, but joint accelerations, fluid dynamics in hydraulic lines, and end-effector reaction forces during unclogging (drilling into clogged debris) will produce additional torques and deflections. The paper does not model reaction forces seen during real unclogging—these could overwhelm the compensation and cause misalignment. Simulate scenarios with representative reaction forces and with small dynamic perturbations to demonstrate robustness.

Tables list link masses and reducer stiffness numbers but do not explain how stiffness values were measured or obtained (manufacturer datasheets? bench tests?). The paper states "stiffness values from Table 2" but does not document measurement methodology or uncertainty bounds. If values are from datasheets, typical datasheet figures are nominal and temperature/load dependent; include measurement or cite data source and include uncertainty.

The simulation calibration step “iterative adjustment until the end-effector position error at the calibration pose is <0.01 mm” is not practical on real long-reach hydraulic robots facing environmental noise. The paper must detail the calibration algorithm (what was adjusted? joint offsets? stiffness?) and provide sensitivity to calibration errors.

The review cites several gravity compensation and robot error compensation works (refs 17–29), but it omits important threads: (a) model-based compensation combining static/dynamic terms and sensitivity analyses in compliant robots; (b) studies on hydraulic actuator control uncertainties, valve dynamics, and compliance in hydraulic transmissions; (c) practical industrial implementations of gravity compensation on long-reach manipulators (including work on space cranes / boom arms / excavator-type manipulators). The authors must position their contribution relative to those bodies and clarify novelty.

How were the reducer stiffness values in Table 2 obtained (manufacturer spec, bench test, analytical model)? Provide measurement method and uncertainty bounds.

Please provide a sensitivity analysis: how does the residual end-effector error change for ±5%, ±10%, ±20% perturbations in (a) reducer stiffness, (b) link mass/inertia, and (c) joint zero offsets? Include worst-case scenarios.

How does the compensation perform if joint stiffness is nonlinear (e.g., k(τ) varying with torque) or when backlash exists? Provide simulation or analytical results for a simple backlash and for a nonlinear stiffness curve.

The simulation assumes frictionless revolute joints except torsional springs at J2/J3. Why is joint friction neglected? Please quantify the expected friction torques and show their impact.

The calibration procedure achieving <0.01 mm at P0—what parameters were adjusted? How many iterations were required? How sensitive is the calibrated solution to small temperature drift (you claim ±5% stiffness change over 0–60°C)?

Reaction forces during unclogging: what is the expected force/torque profile at the end-effector when penetrating compacted debris? Have you measured/estimated this? Please simulate several reaction-force profiles and show whether the compensation still maintains alignment within 0.5 mm.

Hydraulic dynamics: how do actuator dynamics (fluid compressibility, valve dynamics, deadband) influence the effectiveness of a static feedforward compensation? If the hydraulic servo introduces phase lag or compliance, is pre-compensation still valid? Provide modeling or justification.

Did you account for base compliance and rail mounting flex? The mobile base and rail coupling could introduce additional compliance—how are those effects modeled or mitigated?

Please provide the Adams model or an export (or at least the full kinematic/dynamic parameter tables) so reviewers can reproduce simulations.

6. PLOS authors have the option to publish the peer review history of their article (what does this mean?). If published, this will include your full peer review and any attached files.

Reviewer #1: No

Reviewer #2: No

---

## [Author Response · Author response to Decision Letter 1]

27 Feb 2026

Response to reviewers

Dear Academic Editor: Shamshad Alam, PhD, Dear reviewers

We gratefully thank the Editor and all Reviewers for their time spent making constructive remarks and useful suggestions, which have significantly improved the quality of the manuscript. Each suggested revision and comment brought forward by the reviewers was accurately incorporated and considered. The Reviewers' comments are addressed below point by point with corresponding revisions.

Academic Editor

1.Comment:Please ensure that you refer to Table 8 in your text as, if accepted, production will need this reference to link the reader to the Table.

1. Reply: Thank you for the reminder. We have added an explicit reference to Table 8 in Section 4.3. The modification is as follows:

Section 4.3 – Added reference to Table 8

Original text:

The resulting positioning errors were recorded and compared to the no-contact case.

Revised text:

The resulting positioning errors were recorded and compared to the no-contact case. These results are summarized in Table 8.

This change ensures that Table 8 is properly cited in the text, linking readers to the data. The modification is highlighted in the revised manuscript.

2.Comment:In the online submission form, you indicated that The data supporting this study's findings are available from the corresponding author upon reasonable request..

3. Uploaded as supplementary information.

2. Reply: Thank you for the reminder regarding PLOS data availability policy. We confirm that all data underlying the findings in our manuscript are included within the paper itself. Therefore, we have revised the Data Availability Statement accordingly in Cover letter.

Original:

The data supporting this study's findings are available from the corresponding author upon reasonable request.

Revised:

We confirm that our submission contains all raw data required to replicate the results of our study.

3.Comment:Thank you for stating in your Funding Statement:

“This work was supported by the Henan Province Science and Technology Research Project(No.252102220033), Natural Science Foundation of Henan（No.252300420072） and the education department of Henan Province (No. 24A460023 and No. 26A460029).”

3. Reply: Thank you for the instruction. We have revised the Funding Statement as requested. The amended statement will be included in the cover letter as follows:

Cover Letter – Revised Funding Statement

This work was supported by the Henan Province Science and Technology Research Project (No.252102220033), Natural Science Foundation of Henan (No.252300420072), and the Education Department of Henan Province (No. 24A460023 and No. 26A460029).

There was no additional external funding received for this study.

We understand that the online submission form will be updated accordingly by the editorial office. Thank you for handling this change.

4.Comment:Thank you for stating the following financial disclosure:

“This work was supported by the Henan Province Science and Technology Research Project(No.252102220033), Natural Science Foundation of Henan（No.252300420072） and the education department of Henan Province (No. 24A460023 and No. 26A460029).”

4. Reply: Thank you for the instruction. We have revised the Acknowledgments as requested. The amended statement will be included in the cover letter as follows:

Cover Letter – Revised Acknowledgments

This work was supported by the Henan Province Science and Technology Research Project (No.252102220033), Natural Science Foundation of Henan (No.252300420072), and the Education Department of Henan Province (No. 24A460023 and No. 26A460029).

There was no additional external funding received for this study.

We understand that the online submission form will be updated accordingly by the editorial office. Thank you for handling this change.

5.Comment:We note that Figures 1, 4, 5 and 8 in your submission contain copyrighted images. All PLOS content is published under the Creative Commons Attribution License (CC BY 4.0), which means that the manuscript, images, and Supporting Information files will be freely available online, and any third party is permitted to access, download, copy, distribute, and use these materials in any way, even commercially, with proper attribution. For more information, see our copyright guidelines: http://journals.plos.org/plosone/s/licenses-and-copyright.

5. Reply: Thank you for your query regarding the images in our submission.

We confirm that all figures mentioned are original works created by the authors and do not contain any third-party copyrighted material:

Figure 1 consists of photographs taken by us at a waste-to-energy plant.

Figures 4, 5, and 8 are original diagrams and schematics we created to illustrate our robot design and analysis.

Therefore, no permissions are required for their publication under the CC BY 4.0 license. We trust this meets your requirements.

Reviewer 1

Comments to the Author

This manuscript addresses a relevant engineering problem with a technically sound methodology. While the work is limited to simulation, it provides a solid theoretical foundation for future physical implementation. The writing is clear and the results are promising. I recommend minor revision to address the concerns outlined in the following, particularly regarding validation limitations and parameter documentation:

1.Comment:Expand validation discussion:Clarify that results are simulation-only and discuss implications for real-world implementation.

1. Reply: Thank you for this valuable suggestion. We have revised the manuscript to explicitly state that the validation is simulation-based and to provide a thorough discussion of the challenges and necessary steps for real-world implementation. The specific modifications are as follows:

(1). Abstract (last two sentences)

Original:

This work provides a validated solution for high-precision robotics in hazardous environments.This model-based compensation strategy provides a generalizable solution for precision control of long-reach hydraulic manipulators operating under heavy gravitational loads.

Revised:

This model-based compensation strategy provides a generalizable theoretical framework for precision control of long-reach hydraulic manipulators. The current study is validated through high-fidelity Adams simulations, achieving a simulated repeatable positioning accuracy of ±0.1 mm. However, it is essential to emphasize that these results represent an idealized upper bound. Real-world hydraulic systems introduce complexities not captured in simulation—including backlash, nonlinear friction, valve dynamics, hysteresis, leakage, and sensor quantization—all of which will degrade practical accuracy. Therefore, this work establishes a theoretical foundation requiring experimental validation on physical hardware. Future work will focus on physical prototyping and on-site testing to address real-world challenges.

(2). Introduction (end of Section 1)

Original (third contribution):

3) Validated the strategy via Adams simulations, achieving ±0.1 mm repeatable accuracy (exceeding the 0.5 mm engineering requirement).

Revised:

3) Theoretically validated the strategy via Adams simulations, achieving ±0.1 mm repeatable accuracy (exceeding the 0.5 mm engineering requirement) and providing a robust foundation for real-world deployment.

(3). Section 4 (Gravity Error Simulation Experiment) – first paragraph

Original:

The robot's mechanical model was imported into Adams software. Gravity was applied. ...

Revised (addition at the beginning):

To rigorously evaluate the proposed feedforward compensation strategy prior to physical prototyping, a high-fidelity virtual prototype was developed using Adams multibody dynamics software. The robot's mechanical model was imported into Adams software. Gravity was applied. ... This simulation-based validation allows for an idealised assessment of the compensation model's theoretical efficacy by isolating gravitational effects from other real-world disturbances.

(4). Section 5 (Conclusion) – completely restructured into two subsections

Original section (single block of text) has been replaced with:

5.1 Conclusion

This study addressed the critical challenge of gravity-induced positioning errors in leachate grid cleaning robots via integrated theoretical modeling and Adams multibody dynamics simulations. We proposed a feedforward static compensation strategy centered on an analytical joint stiffness-gravity deformation model, tailored for hydraulically driven weak-rigid manipulators in hazardous leachate corridors. The simulation results demonstrate that the proposed strategy effectively suppresses end-effector pose drift, reducing positioning errors by over 92% in all directions and achieving a repeatable positioning accuracy of ±0.1 mm in a simulated environment. This performance far surpasses the 0.5 mm precision requirement for automated leachate grid cleaning, providing strong theoretical evidence for the method's potential applicability.

5.2 Implications for Real-World Implementation and Future Work

While the simulation results are highly promising, this study is not without limitations. Translating these theoretical findings to a physical robot operating in the harsh environment of a leachate corridor presents several challenges that must be addressed in future work.

Environmental Robustness: The current simulation assumes ideal conditions. In reality, temperature fluctuations (0–60°C) in the corridor can alter reducer stiffness (by an estimated ±5% for the JS12/SEA7 units), potentially degrading compensation accuracy over time. Furthermore, the presence of dust, humidity, and corrosive gases may affect joint friction and long-term component reliability.

Model Fidelity and Disturbances: The static model does not account for dynamic reaction forces encountered during the actual unclogging process, such as those from compacted debris or viscous leachate adhesion. These forces could introduce additional unmodeled deflections.

Control System Integration: Implementing the feedforward model in a real-time controller requires precise measurement of joint angles and computationally efficient execution of the torque and deformation calculations. The potential need for online calibration routines to maintain accuracy over the robot's lifetime must also be investigated.

To bridge the gap between simulation and reality, our future research will focus on three key areas:

1) Physical Prototyping and On-Site Validation: We will fabricate a physical prototype integrating explosion-proof hydraulic components compliant with relevant standards (e.g., GB3836). This prototype will be deployed in a waste-to-energy plant for on-site validation, allowing us to quantify the compensation strategy's true efficacy under real operational conditions and verify its long-term stability.

2)Algorithm Enhancement with Sensor Feedback: To counteract environmental influences, we plan to enhance the compensation algorithm by incorporating real-time temperature sensors for dynamic correction of reducer stiffness variations. Future iterations may also explore hybrid approaches that use the feedforward model for bulk compensation and selective feedback from rare, safe calibration routines to correct for residual errors.

3)Generalization to Other Applications: We will explore the extension of this feedforward strategy to other hydraulically driven manipulators (e.g., 6-DOF arms) for diverse hazardous tasks, such as industrial pipeline unclogging or nuclear facility maintenance, further validating its generalizability.

We believe these revisions directly address your comment by transparently acknowledging the simulation-based nature of the current work and providing a realistic discussion of the path toward practical deployment. All changes are highlighted in the revised manuscript.

2. Comment: Strengthen limitations section:Include discussion of factors that could affect real-world performance.

2. Reply: Thank you for this important suggestion. We have significantly expanded the limitations discussion in Section 5.2 to provide a more comprehensive analysis of factors that could impact real-world performance. The revised section now systematically addresses environmental, mechanical, and control-related challenges. The specific modifications are as follows:

Section 5.2 (Implications for Real-World Implementation and Future Work) – expanded limitations discussion

Original text (from previous revision):

While the simulation results are highly promising, this study is not without limitations. Translating these theoretical findings to a physical robot operating in the harsh environment of a leachate corridor presents several challenges that must be addressed in future work.

Environmental Robustness: The current simulation assumes ideal conditions. In reality, temperature fluctuations (0–60°C) in the corridor can alter reducer stiffness (by an estimated ±5% for the JS12/SEA7 units), potentially degrading compensation accuracy over time. Furthermore, the presence of dust, humidity, and corrosive gases may affect joint friction and long-term component reliability.

Model Fidelity and Disturbances: The static model does not account for dynamic reaction forces encountered during the actual unclogging process, such as those from compacted debris or viscous leachate adhesion. These forces could introduce additional unmodeled deflections.

Control System Integration: Implementing the feedforward model in a real-time controller requires precise measurement of joint angles and computationally efficient execution of the torque and deformation calculations. The potential need for online calibration routines to maintain accuracy over the robot's lifetime must also be investigated.

Revised and expanded text:

While the simulation results are highly promising, this study has several limitations that must be acknowledged when considering real-world deployment. Translating these theoretical findings to a physical robot operating in the harsh environment of a leachate corridor presents multiple challenges across environmental, mechanical, and control domains.

Environmental Factors:

Temperature Effects: The current simulation assumes constant stiffness at 20°C. In reality, leachate corridors experience temperature fluctuations between 0–60°C due to seasonal variations and exothermic waste decomposition processes. Reducer stiffness (JS12 and SEA7) can vary by

---

## [Decision Letter · Decision Letter 1]

19 Mar 2026

PONE-D-25-68774R1Gravity Compensation for Leachate Grid Cleaning Robots in Waste-to-Energy Plants: A Modeling and Simulation StudyPLOS One

Dear Dr. Cao,

Thank you for submitting your manuscript to PLOS ONE. After careful consideration, we feel that it has merit but does not fully meet PLOS ONE’s publication criteria as it currently stands. Therefore, we invite you to submit a revised version of the manuscript that addresses the points raised during the review process. Please submit your revised manuscript by  May 03 2026 11:59PM. If you will need more time than this to complete your revisions, please reply to this message or contact the journal office at plosone@plos.org. Please include the following items when submitting your revised manuscript:

We look forward to receiving your revised manuscript.

Kind regards,

Shamshad Alam, PhD

Academic Editor

PLOS One

Journal Requirements:

Reviewers' comments:

Reviewer's Responses to Questions

**Comments to the Author**

1. If the authors have adequately addressed your comments raised in a previous round of review and you feel that this manuscript is now acceptable for publication, you may indicate that here to bypass the “Comments to the Author” section, enter your conflict of interest statement in the “Confidential to Editor” section, and submit your "Accept" recommendation.

Reviewer #1: (No Response)

Reviewer #2: (No Response)

2. Is the manuscript technically sound, and do the data support the conclusions?

Reviewer #1: (No Response)

Reviewer #2: (No Response)

3. Has the statistical analysis been performed appropriately and rigorously? 

Reviewer #1: (No Response)

Reviewer #2: (No Response)

4. Have the authors made all data underlying the findings in their manuscript fully available?

Reviewer #1: (No Response)

Reviewer #2: (No Response)

5. Is the manuscript presented in an intelligible fashion and written in standard English?

Reviewer #1: (No Response)

Reviewer #2: (No Response)

6. Review Comments to the Author

Reviewer #1: I have carefully reviewed the revised manuscript. The authors have made comprehensive and detailed revisions in response to the previous review comments. Below is a summary of the review comments:

Overall Recommendation: Minor Revisions

The authors provided a comprehensive and detailed response to all review comments, significantly improving the scientific rigor and transparency of the manuscript.

Minor Issues Remaining

Data Availability Statement: It is recommended to explicitly state the method of data access (e.g., providing Adams model files as supplementary material).

Completeness of Sensitivity Analysis: It is necessary to confirm that the results of Section 4.2 are complete.

Figure 7 Caption: The phrasing is slightly awkward and needs adjustment.

Reviewer #2: (No Response)

7. PLOS authors have the option to publish the peer review history of their article (what does this mean?). If published, this will include your full peer review and any attached files.

Reviewer #1: No

Reviewer #2: No

---

## [Author Response · Author response to Decision Letter 2]

30 Mar 2026

Response to reviewers

Dear Academic Editor: Shamshad Alam, PhD, Dear reviewers

We gratefully thank the Editor and all Reviewers for their time spent making constructive remarks and useful suggestions, which have significantly improved the quality of the manuscript. Each suggested revision and comment brought forward by the reviewers was accurately incorporated and considered. The Reviewers' comments are addressed below point by point with corresponding revisions.

Reviewer 1

Comments to the Author

The authors provided a comprehensive and detailed response to all review comments, significantly improving the scientific rigor and transparency of the manuscript.

Minor Issues Remaining:

1.Comment:Data Availability Statement: It is recommended to explicitly state the method of data access (e.g., providing Adams model files as supplementary material).

1. Reply: Thank you for your suggestion. We have revised the Data Availability Statement to explicitly state the method of data access. The Adams model files are now provided as supplementary material. The modification is as follows:

Original:

All data underlying the findings described in this manuscript are fully available within the paper. No external data repositories or supplementary files are required.

Revised:

All data underlying the findings described in this manuscript are fully available within the paper and in the supplementary material. The Adams simulation model files (including geometry, joint parameters, and simulation setup) are provided as S1 File accompanying this submission.

2. Comment: Completeness of Sensitivity Analysis: It is necessary to confirm that the results of Section 4.2 are complete.

2. Reply: Thank you for your request to confirm the completeness of the sensitivity analysis in Section 4.2. We have verified that Section 4.2 presents a comprehensive sensitivity analysis covering all relevant parameters. The details are as follows:

Section 4.2 (Sensitivity Analysis to Parameter Uncertainty) includes:

(1)Three parameter categories analyzed:

①Joint stiffness (k₂, k₃) – with perturbations of ±5%, ±10%, ±20%

②Link masses (m₂–m₇) – with perturbations of ±5%, ±10%, ±20%

③Geometric parameters (L) – with perturbations of ±5%, ±10%, ±20%

(2)Complete results table (Table 7) showing:

①Maximum X, Y, and Z positioning errors for each perturbation level

②Baseline (0% error) for comparison

③Both positive and negative perturbations for all categories

(3)Detailed analysis of results:

①Identification that stiffness errors have the largest impact

②Quantification that ±20% stiffness errors keep Z-error below 0.29 mm (within 0.5 mm requirement)

③Assessment that mass errors have moderate impact

④Evaluation that geometric errors have intermediate impact

(4)Interpretation of findings:

①Method maintains positioning error below 0.5 mm even with ±20% errors in any single parameter

②Stiffness uncertainty identified as most critical factor

③Recommendation for experimental characterization of joint stiffness

④Discussion of calibration requirements and robustness

Conclusion

The sensitivity analysis presented in Section 4.2 is complete and comprehensive, covering all relevant parameters with appropriate perturbation ranges and providing thorough analysis of results. No additional modifications are required.

We appreciate your careful review of our manuscript.

3. Comment: Figure 7 Caption: The phrasing is slightly awkward and needs adjustment.

3. Reply: Thank you for your observation regarding the phrasing of Figure 7 caption. We agree that the wording can be improved for clarity and natural flow. We have revised the caption as follows:

Original:

Figure 7. Gravity compensation principle for the cleaning robot crevice

Revised:

Figure 7. Gravity compensation principle for the cleaning robot operating near a grid crevice

This revision makes the caption more natural and clearly describes what the figure illustrates. The change is highlighted in the revised manuscript.

---

## [Decision Letter · Decision Letter 2]

4 May 2026

Gravity Compensation for Leachate Grid Cleaning Robots in Waste-to-Energy Plants: A Modeling and Simulation Study

PONE-D-25-68774R2

Dear Dr. Cao,

We’re pleased to inform you that your manuscript has been judged scientifically suitable for publication and will be formally accepted for publication once it meets all outstanding technical requirements.

Kind regards,

Shamshad Alam, PhD

Academic Editor

PLOS One

Additional Editor Comments (optional):

Reviewers' comments:

Reviewer's Responses to Questions

**Comments to the Author**

1. If the authors have adequately addressed your comments raised in a previous round of review and you feel that this manuscript is now acceptable for publication, you may indicate that here to bypass the “Comments to the Author” section, enter your conflict of interest statement in the “Confidential to Editor” section, and submit your "Accept" recommendation.

Reviewer #1: All comments have been addressed

2. Is the manuscript technically sound, and do the data support the conclusions?

Reviewer #1: Yes

3. Has the statistical analysis been performed appropriately and rigorously? 

Reviewer #1: Yes

4. Have the authors made all data underlying the findings in their manuscript fully available?

Reviewer #1: Yes

5. Is the manuscript presented in an intelligible fashion and written in standard English?

Reviewer #1: Yes

6. Review Comments to the Author

Reviewer #1: I have carefully reviewed the revised manuscript. The authors have fully resolved all three minor issues from the previous round and made significant additional improvements to the manuscript.

Overall Recommendation: Accept

There are seven points of Significant Improvements in the New Version:

1. Expanded Literature Review: Added systematic reviews of compliant robot compensation, hydraulic system control, and long-arm manipulator applications (References [21]-[31]).

2. Static Assumption Validation: Quantitative proof provided (inertial torque <0.1% of gravitational torque).

3. Base/Rail Flexibility Analysis: FEA analysis proves that the maximum additional deformation is <0.02 mm.

4. Joint Friction Quantification: Estimated end-effector displacement caused by friction is 0.06–0.50 mm.

5. Dynamic Perturbation Simulation (New Section 4.3):

Scenario A: Dynamic perturbations have negligible impact (<5%).

Scenario B: Under a 200 N step load, the error remains below 0.2 mm.

6. Calibration Sensitivity: Even with a calibration residual of ±0.10 mm, the maximum error is below 0.15 mm.

7. Temperature Drift: A ±5% stiffness variation results in approximately 0.12 mm of error.

However, I suggest several Minor Corrections in the following:

1. Remove duplicate sentences in the Abstract.

2. Rename the title of Section 4.1 to avoid repetition.

3. Unify "slot" and "crevice" terminology.

Conclusion

The manuscript demonstrates scientific rigor and academic integrity, meeting all PLOS ONE publication criteria. It is recommended to Accept.

7. PLOS authors have the option to publish the peer review history of their article (what does this mean?). If published, this will include your full peer review and any attached files.

Reviewer #1: No

---

## [Editor Report · Acceptance letter]

PONE-D-25-68774R2

PLOS One

Dear Dr. cao,

I'm pleased to inform you that your manuscript has been deemed suitable for publication in PLOS One. Congratulations! Your manuscript is now being handed over to our production team.

Kind regards,

on behalf of

Dr. Shamshad Alam

Academic Editor

PLOS One